# Variation in human herpesvirus 6B telomeric integration, excision, and transmission between tissues and individuals

Michael L Wood[1], Colin D Veal[1], Rita Neumann[1], Nicolás M Suárez[2], Jenna Nichols[2], Andrei J Parker[1], Diana Martin[1], Simon PR Romaine[3,4], Veryan Codd[3], Nilesh J Samani[3], Adriaan A Voors[5], Maciej Tomaszewski[6], Louis Flamand[7], Andrew J Davison[2], Nicola J Royle[1]*

[1]Department of Genetics and Genome Biology, University of Leicester, Leicester, United Kingdom; [2]MRC-University of Glasgow Centre for Virus Research, Glasgow, United Kingdom; [3]Department of Cardiovascular Sciences, University of Leicester, Leicester, United Kingdom; [4]NIHR Leicester Biomedical Research Centre, Glenfield Hospital, Leicester, United Kingdom; [5]University of Groningen, Department of Cardiology, University Medical Center Groningen, Groningen, Netherlands; [6]Division of Cardiovascular Sciences, School of Medical Sciences, Faculty of Biology, Medicine and Health, University of Manchester, Manchester, United Kingdom; [7]Department of Microbiology, Infectious Diseases and Immunology, Faculty of Medicine, Université Laval, Quebec City, Québec, Canada

*For correspondence:
njr@le.ac.uk

**Abstract** Human herpesviruses 6A and 6B (HHV-6A/6B) are ubiquitous pathogens that persist lifelong in latent form and can cause severe conditions upon reactivation. They are spread by community-acquired infection of free virus (acqHHV6A/6B) and by germline transmission of inherited chromosomally integrated HHV-6A/6B (iciHHV-6A/6B) in telomeres. We exploited a hypervariable region of the HHV-6B genome to investigate the relationship between acquired and inherited virus and revealed predominantly maternal transmission of acqHHV-6B in families. Remarkably, we demonstrate that some copies of acqHHV-6B in saliva from healthy adults gained a telomere, indicative of integration and latency, and that the frequency of viral genome excision from telomeres in iciHHV-6B carriers is surprisingly high and varies between tissues. In addition, newly formed short telomeres generated by partial viral genome release are frequently lengthened, particularly in telomerase-expressing pluripotent cells. Consequently, iciHHV-6B carriers are mosaic for different iciHHV-6B structures, including circular extra-chromosomal forms that have the potential to reactivate. Finally, we show transmission of an HHV-6B strain from an iciHHV-6B mother to her non-iciHHV-6B son. Altogether, we demonstrate that iciHHV-6B can readily transition between telomere-integrated and free virus forms.

## Introduction

Human herpesviruses 6A and 6B (HHV-6A/6B; species *Human betaherpesvirus 6A* and *Human beta-herpesvirus 6B*) are closely related viruses, sharing approximately 90% sequence identity (*Ablashi et al., 2014*). Their genomes comprise a unique region (U) of approximately 140 kb flanked on each side by a direct repeat (DR) of approximately 8 kb (DR$_L$ on the left and DR$_R$ on the right). A tandem repeat reiteration is located near each end of DR. T1 at one end of DR consists of 1–8 kb of telomere

(TTAGGG) and degenerate telomere-like repeats (*Huang et al., 2014*; *Lindquester and Pellett, 1991*; *Zhang et al., 2017*). T2 at the other end of DR is much shorter and contains only (TTAGGG) repeats (*Achour et al., 2009*; *Zhang et al., 2017*). HHV-6A/6B have the capacity to integrate into human telomeres, most likely through a homology-dependent recombination mechanism facilitated by the perfect telomere repeats in T2 (*Arbuckle et al., 2010*; *Wallaschek et al., 2016b*). The precise integration mechanism has not been defined, and searches for regulatory viral or host factors are underway (*Gilbert-Girard et al., 2017*; *Gilbert-Girard et al., 2020*; *Wallaschek et al., 2016a*; *Wight et al., 2018*).

HHV-6A/6B infect over 90% of the global population in early childhood; this usually manifests with mild symptoms but nevertheless often requires medical attention (*Asano et al., 1994*; *Hall et al., 1994*; *Kondo et al., 1993*; *Ward and Gray, 1994*; *Zerr et al., 2005*). As with most herpesviruses, HHV-6A/6B enter lifelong latency following primary infection, and the most serious impacts on health appear to occur when the virus reactivates. Reactivation of free virus acquired in the community (acqHHV-6A/6B) has been shown to cause viral encephalitis, drug-induced hypersensitivity syndrome/ drug reaction with eosinophilia and systemic symptoms (DIHS/DRESS), and acute graft-versus-host disease (aGVHD) (*Aihara et al., 2003*; *Eshki et al., 2009*; *Pritchett et al., 2012*; *Prusty, 2018*; *Yao et al., 2010*; *Yoshikawa et al., 2006*). In addition, HHV-6A infection has been associated with multiple sclerosis and other chronic neurological conditions (*Yao et al., 2010*). Herpesviruses typically achieve latency by forming circular DNA episomes within specific cell types, but HHV-6A/6B episomes have not been detected and it has been proposed that latency of these viruses is achieved through telomeric integration (*Arbuckle et al., 2010*).

As a result of historical germline integrations into telomeres, approximately 1% of the population carries inherited chromosomally integrated HHV-6A/6B (iciHHV-6A/6B), usually with a full-length copy of the viral genome in every cell (*Arbuckle et al., 2010*; *Daibata et al., 1999*; *Huang et al., 2014*; *Leong et al., 2007*; *Morris et al., 1999*; *Tanaka-Taya et al., 2004*). HHV-6A/6B appear to have the potential to integrate into any telomere; however, most contemporary instances of iciHHV-6A/6B are derived from a relatively small number of ancient integrations in a limited number of chromosome ends (*Aswad et al., 2021*; *Greninger et al., 2018*; *Kawamura et al., 2017*; *Liu et al., 2018*; *Liu et al., 2020*; *Miura et al., 2018*; *Tweedy et al., 2016*; *Zhang et al., 2017*). For example, iciHHV-6B is commonly found integrated into the telomere on the short arm of chromosome 17 (17p) in populations of European descent and iciHHV-6A is commonly found in the long arm of chromosome 22 (22q) in Japan. Integration sites have been determined in a variety of studies by one or a combination of the following methods: fluorescent in situ hybridization (FISH), PCR amplification from subterminal sequences adjacent to human telomeres (subtelomere) into the viral genome, inverse PCR and comparison with known subtelomere sequences, long-read sequencing, and optical genome mapping (*Daibata et al., 1999*; *Huang et al., 2014*; *Luppi et al., 1998*; *Nacheva et al., 2008*); (*Aswad et al., 2021*; *Greninger et al., 2018*; *Kawamura et al., 2017*; *Liu et al., 2020*; *Miura et al., 2018*; *Tweedy et al., 2016*; *Wight et al., 2020*; *Zhang et al., 2017*).

Detection of viral gene expression in some iciHHV-6A/6B carriers has raised the possibility of a deleterious impact on the individual's health over their lifetime (*Huang et al., 2014*; *Peddu et al., 2019*; *Strenger et al., 2014*). Similarly, the presence of a large viral genome within a telomere has been shown to cause localized instability that may influence telomere DNA damage signaling and function (*Huang et al., 2014*; *Wood and Royle, 2017*; *Zhang et al., 2016*). Recent studies have also indicated an association between iciHHV-6A/6B and angina pectoris (*Gravel et al., 2015*); unexplained infertility (*Miura et al., 2020*); an increased risk of aGVHD following hematopoietic stem cell transplantation when either the donor or recipient has iciHHV-6A/6B (*Hill et al., 2017*; *Weschke et al., 2020*); complications following solid organ transplantation when there was iciHHV-6A/6B donor/recipient mismatch (*Bonnafous et al., 2018*; *Bonnafous et al., 2020*; *Petit et al., 2020*); and possibly an increased risk of pre-eclampsia (*Gaccioli et al., 2020*).

There is increasing evidence that iciHHV-6A/6B genomes can reactivate fully (*Endo et al., 2014*) and then be transmitted as acqHHV-6A/6B (*Gravel et al., 2013*; *Hall et al., 2010*), but the chain of events leading to reactivation is unknown. Telomeres are terminated by a single-stranded 3′ overhang that can strand-invade into the upstream double-stranded telomeric DNA to form a telomere loop (t-loop) (*Doksani et al., 2013*; *Griffith et al., 1999*; *Wang et al., 2004*). This secondary structure is stabilized by TRF2, a component of the Shelterin complex that binds to double-stranded telomere

repeats, and plays a protective role by preventing the ends of linear chromosomes being detected as double-strand breaks and inappropriately repaired (*de Lange, 2018*; *Schmutz et al., 2017*; *Stansel et al., 2001*; *Van Ly et al., 2018*). However, t-loops can be excised as telomere repeat-containing t-circles (*Pickett et al., 2009*; *Pickett et al., 2011*; *Sarek et al., 2015*; *Tomaska et al., 2019*) by the SLX1-4 complex, which is a structure-specific endonuclease (*Fekairi, 2009*; *Vannier et al., 2012*; *Wan et al., 2013*), and unregulated excision of t-loops results in critically short telomeres (*Deng et al., 2013*; *Pickett et al., 2011*; *Rivera et al., 2017*). We and others have proposed that release of the viral genome is a prerequisite for reactivation of iciHHV-6A/6B and that this release is driven by normal t-loop processing (*Huang et al., 2014*; *Prusty et al., 2013*; *Wood and Royle, 2017*).

Here, we have investigated iciHHV-6A/6B genomes (with a particular focus on iciHHV-6B), their associated telomeres, and their relationships to acqHHV-6B genomes in families and communities. We exploited the proximal hypervariable region of T1 (pvT1) in the HHV-6B genome to distinguish between acqHHV-6B strains and predict relationships between iciHHV-6B and acqHHV-6B in families. For the first time, we have detected acqHHV-6B telomeric integration in saliva DNA from healthy adults. The frequency of partial or complete iciHHV-6B excision varied between germline and somatic cells, with notably high frequencies in two pluripotent cell lines, and we present evidence of HHV-6B transmission from an iciHHV-6B carrier mother to her non-iciHHV-6B son. In summary, our study demonstrates that HHV-6B can readily transition between iciHHV-6B and acqHHV-6B forms, and vice versa.

## Results
### DR$_R$-pvT1 tracking of HHV-6B transmission

To explore the relationship between iciHHV-6B and acqHHV-6B, we identified a genomic marker (pvT1) capable of distinguishing viral strains by nested PCR and sequencing. To investigate DR$_R$-pvT1 specifically, the first round of amplification was achieved using a primer anchored at the end of U near DR$_R$ and a primer in DR on the other side of T1 (*Figure 1A*). The second round of amplification and subsequent sequencing involved a primer (TJ1F) anchored in a short, conserved sequence approximately 400 bases into the T1 telomere-like repeat array and a second primer in DR (DR421R), adjacent to T1. A total of 102 DNA samples were analyzed: iciHHV-6B from 39 individuals (38.2%) and acqHHV-6B from 63 (61.8%) individuals (*Supplementary file 1*). The analysis showed that DR$_R$-pvT1 can be divided into three regions from the DR421 primer (proximal, central, and distal), with most of the variation being due to differences in the number and distribution of telomere (TTAGGG) and degenerate telomere-like repeats (predominantly CTATGG and CTAGGG) in the proximal and central regions. Patterns of repeats are shown for a subset of samples in *Figure 1B*, and the complete dataset is provided in *Figure 1—figure supplement 1*.

Among the 102 repeat patterns, 93 were different from each other, and 90 of these were detected in only a single donor or family. Amongst the 12 repeat patterns shared between individuals, 8 were identical to each other, and 7 of these were from closely related viral genome sequences in iciHHV-6B carriers with a shared 9q integration. The eighth was found in saliva DNA from an individual (SAL030) with acqHHV-6B (*Figure 1—figure supplement 1*). Among the remaining four repeat patterns, two were in iciHHV-6B samples (NWA008 and DER512) that share the same 17p integration site, and two were in acqHHV-6B samples (SAL023 and TEL-FA G1P1). Remarkably, the pvT1 repeat pattern was different in almost every unrelated individual with acqHHV-6B (61/63, 96.8%) and also in the majority of iciHHV-6B individuals (30/39, 77.0% ; 29/31, 93.5%, when the entire 9q iciHHV-6B group was excluded).

Although DR$_R$-pvT1 is highly variable among acqHHV-6B genomes in unrelated individuals, this is not the case within families (*Figure 1C*, *Figure 1—figure supplement 2*). Saliva DNA was analyzed from 36 individuals in eight families (without iciHHV-6B) and DR$_R$-pvT1 was successfully amplified and sequenced from 34 of them (94%). 18 of the 20 children (90%) in these families had the same repeat pattern as one of their parents, presumably as a result of acqHHV-6B transmission directly from a parent (maternal transmission in six of the eight families and paternal transmission in one) or indirectly via a sibling (*Figure 1—figure supplement 2*). In contrast, the analysis revealed that one child in each of the two families must have contracted acqHHV-6B from outside their immediate family (TEL-FC G2P2 and TEL-FM G2P1, *Figure 1—figure supplement 2*). As a control, analysis of DR$_R$-pvT1

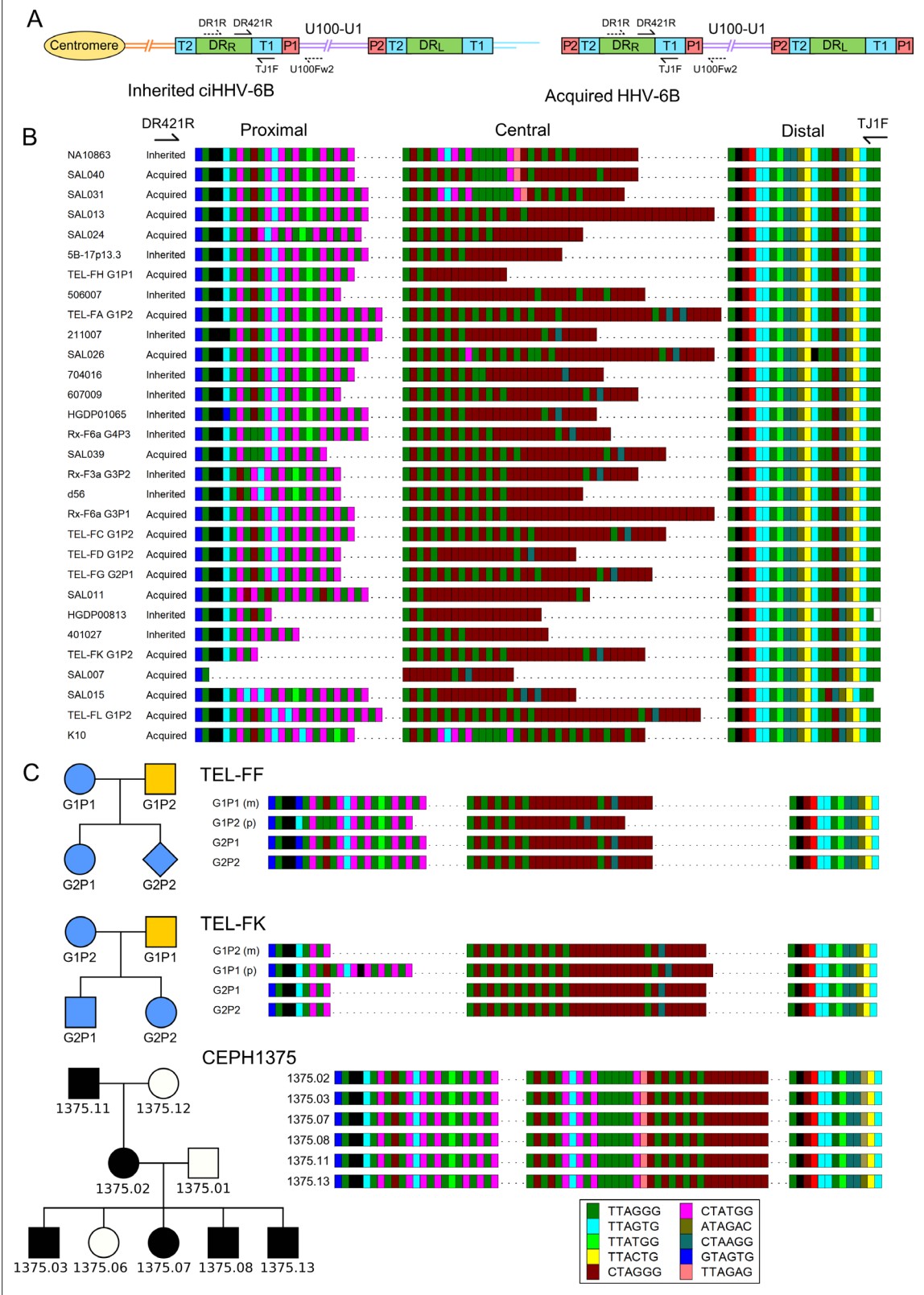

**Figure 1.** Characterization of the highly variable DR$_R$-pvT1 in iciHHV-6B and acqHHV-6B genomes. (**A**) Diagram showing the locations of PCR primers used to amplify the DR$_R$-T1 region specifically (U100Fw2 and DR1R) and the nested primers used to reamplify and sequence pvT1 (DR421R and TJ1F) in the iciHHV-6B genome (left) and in the acqHHV-6B genome (right). (**B**) DR$_R$-pvT1 repeat patterns are shown to demonstrate diversity among a subset of iciHHV-6B (inherited) and acqHHV-6B (acquired) genomes from various individuals. Telomere-like and degenerate repeats present in the HHV-6B

*Figure 1 continued on next page*

*Figure 1 continued*

pvT1 region are color-coded as shown in the key and as follows: dark green, TTAGGG; cyan, TTAGTG; yellow, TTACTG; lime green, TTATGG; brown, CTAGGG; pink, CTATGG; dark yellow, ATAGAC; teal, CTAAGG; blue, GTAGTG; peach, TTAGAG; red, GTCTGG. Black squares represent other less common degenerate repeats, white squares show where the sequence could not be determined accurately. Dashes between repeats were added to maximize alignment and to allow comparison between the sections of DR$_R$-pvT1, labeled proximal, central (highly variable), and distal (highly conserved) with respect to the location of the DR421R primer. (**C**) DR$_R$-pvT1 repeat patterns from acqHHV-6B in two families. None of the family members were iciHHV-6 carriers. In both families, the HHV-6B DR$_R$-pvT1 region in the two children was the same as one parent, indicating that the children carried the same strain of the virus, whereas the second parent carried a different strain. Blue shapes indicate that the child had acqHHV-6B with the same pvT1 repeat map as the mother. Analysis of DR$_R$-pvT1 in the CEPH1375 family showed stable inheritance of the same pvT1 repeat pattern across three generations. iciHHV-6B carriers in the CEPH1375 family are shown as filled black symbols.

The online version of this article includes the following figure supplement(s) for figure 1:

**Figure supplement 1.** Complete set of pvT1 interspersion maps from DR$_R$ in iciHHV-6B (inherited) and acquired HHV-6B (non-inherited) strains.

**Figure supplement 2.** HHV-6B strain identification using DR$_R$-pvT1 repeat patterns in eight families suggests that transmission of acqHHV-6B is predominantly from parents.

---

repeat patterns in iciHHV-6B carriers in the CEPH1375 family demonstrated the stability of the repeat patterns across three generations (*Figure 1C*). All six individuals, shown previously to be iciHHV-6B carriers (*Huang et al., 2014*), shared an identical pattern, indicating that DR$_R$-pvT1 in iciHHV-6B does not necessarily acquire mutations from one generation to the next. In summary, DR$_R$-pvT1 analysis was used successfully to distinguish HHV-6B strains and monitor transmission of iciHHV-6B and acqHHV-6B between family members.

## Phylogeny of iciHHV-6A/6B in relation to integration events

Progress has been made recently in sequencing iciHHV-6A/6B and acqHHV-6A/6B genomes by generating overlapping PCR amplicons (*Zhang et al., 2017*) or an oligonucleotide hybridization enrichment process (*Aswad et al., 2021*; *Greninger et al., 2018*; *Telford et al., 2018*). However, the evolutionary histories of iciHHV-6A/6B genomes and their relationships with acqHHV-6A/6B genomes remain unclear. Moreover, it is not known how many founder germline integration events have occurred. To expand the data required to answer such questions, we screened various cohorts and families to identify additional iciHHV-6A/6B-positive samples, and then selected several key samples for viral genome sequencing using the overlapping PCR amplicon approach (*Supplementary file 1*; *Huang et al., 2014*; *Zhang et al., 2017*). These samples included five iciHHV-6A and four iciHHV-6B samples from the BIOSTAT-CHF cohort, four other iciHHV-6A samples, one iciHHV-6A cell line (HGDP00628 *Huang et al., 2014*), and a pluripotent cell line (d37) from an iciHHV-6B carrier. In total, 10 iciHHV-6A and 5 iciHHV-6B genomes were sequenced.

Distance-based phylogenetic networks of iciHHV-6A/6B and acqHHV-6A/6B genomes were generated, including the 15 newly sequenced iciHHV-6A/6B genomes and iciHHV-6A/6B genomes representing previously identified clades and integration sites (*Figure 2*, *Supplementary file 1*). These networks reinforce previous findings (*Aswad et al., 2021*; *Greninger et al., 2018*; *Tweedy et al., 2016*; *Zhang et al., 2017*), showing that the majority of iciHHV-6A genomes in European and North American carriers (including the 10 newly sequenced) originate from three integration events at 17p , 18q, and 19q, with the time to the most recent common ancestor estimated at 23–105 thousand years ago (*Figure 2A*, *Supplementary file 2*). The networks also confirm that iciHHV-6B carriers with European and North America ancestry originate from a larger number of integration events that occurred more recently (21–25 thousand years ago; *Supplementary file 2*; *Aswad et al., 2021*; *Greninger et al., 2018*; *Zhang et al., 2017*). Inclusion of the newly sequenced iciHHV-6B genomes added two new iciHHV-6B clades to the phylogenetic network, each defined by a distinct integration site (*Figure 2B*). As a consequence of the complex evolution of subterminal sequences in the human genome (*Riethman, 2008*), the chromosomal locations in these iciHHV-6B clades will require further verification by a different method, but they represent two more independent germline telomeric integration events.

As an aid to analyzing the increasing number of HHV-6A/6B genome sequences, we developed HHV-6 Explorer (https://www.hhv6explorer.org/), which is an online interface for monitoring clade-specific variation in DNA and predicted protein sequences. An analysis of sequence variation in relation to integration site using this tool produced new insights, for example, by demonstrating that two

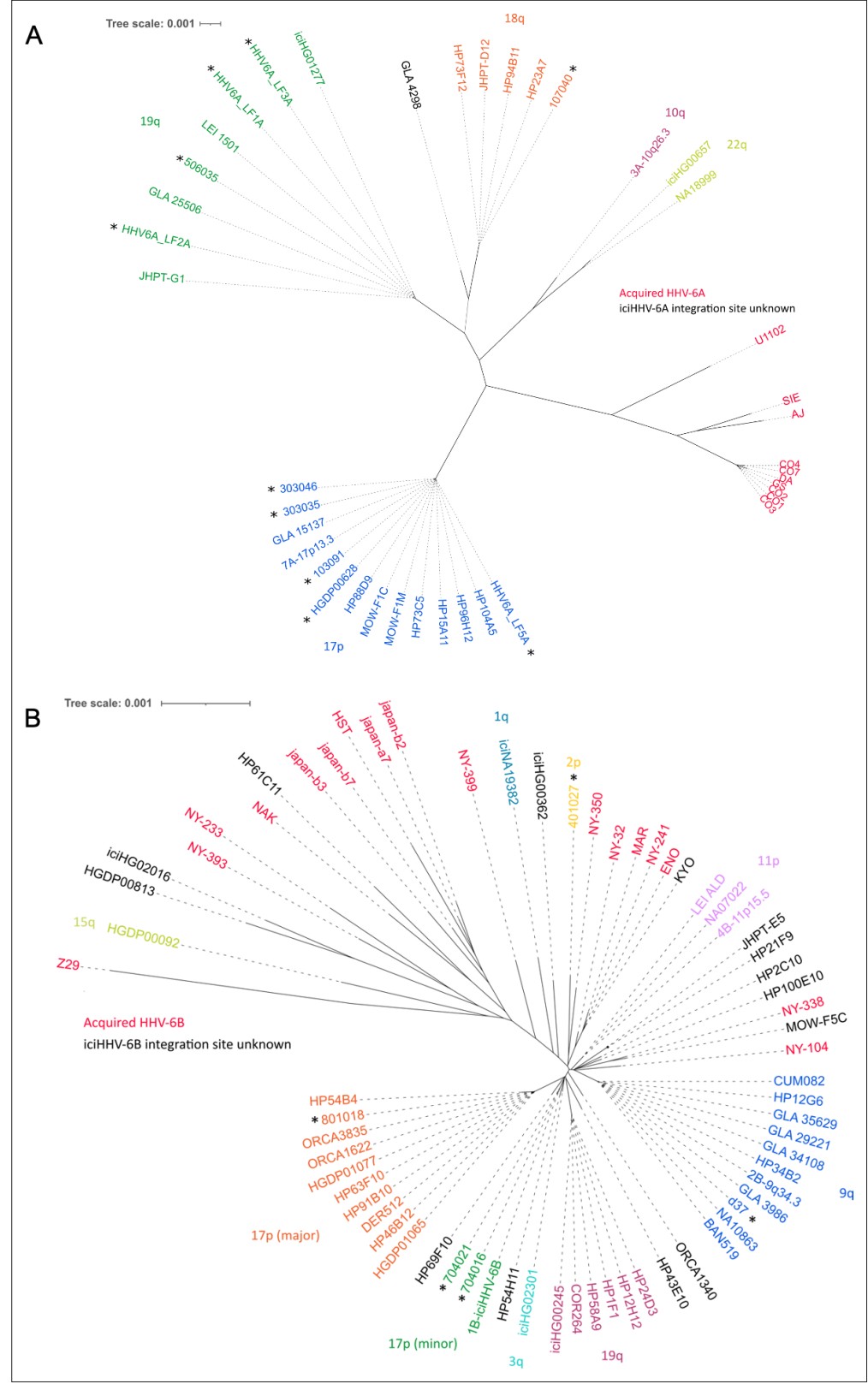

**Figure 2.** Distance-based phylogenetic networks for HHV-6A and HHV-6B. (**A**) A phylogenetic network of 41 HHV-6A viral genome sequences. High-sequence homology to iciHHV-6A genomes for which the integration site had already been established allowed all but one (GLA_4298) iciHHV-6A genomes to be assigned to a telomeric integration site. The colors indicate a chromosomal integration as follows: green, 19q; orange, 18q; purple, 10q;

*Figure 2 continued on next page*

*Figure 2 continued*

yellow-green, 22q; blue, 17p . The name in black identifies the iciHHV-6A strain without a predicted chromosomal location. Non-inherited, acqHHV-6A strains are shown in red. iciHHV-6A genomes sequenced as part of this study are identified by a black asterisk. (**B**) A phylogenetic tree of 68 HHV-6B genome sequences (51 iciHHV-6B and 17 acqHHV-6B). iciHHV-6B genomes sequenced as part of this study are identified by a black asterisk. High-sequence homology to iciHHV-6B genomes with a known integration site (already established by fluorescent in situ hybridization [FISH]) allowed the majority of iciHHV-6B genomes to be assigned a telomeric chromosomal location. From the newly sequenced iciHHV-6B genomes, two new clades were identified and provisionally labeled as: 17p  (minor) and 2p . The 17 p (minor) clade includes the 1B-iciHHV-6B genome, previously a singleton in HHV-6B networks, and the iciHHV-6B genomes in 704021 and 704016. The subtelomere-iciHHV-6B junction sequences, amplified with the 17p311 and DR8F(A/B) primers, are similar for all three samples and distinct form the 17p (major) clade. The other new clade is characterized by the iciHHV-6B genome (401027). This group is provisionally labeled 2p  because the subtelomere-iciHHV-6B junction was amplified by the 2p2 and DR8F(A/B) primers in 401027, and two other samples (410005 and 308006, see *Figure 3C*).

The online version of this article includes the following figure supplement(s) for figure 2:

**Figure supplement 1.** Examples of views taken from the HHV-6 Explorer.

potentially inactivating mutations, U14 in an iciHHV-6B clade and in U79 in an iciHHV-6A clade, must have arisen after integration (*Figure 2—figure supplement 1*).

## Predicting chromosomal iciHHV-6B integration sites from DR$_R$-pvT1 repeat patterns

To explore whether DR$_R$-pvT1 repeat patterns reflect the phylogenetic relationships between iciH-HV-6B genomes, 19 for which DNA was available were subjected to DR$_R$-pvT1 analysis. The results show that iciHHV-6B genomes with a high degree of overall sequence similarity also shared similar DR$_R$-pvT1 repeat patterns (*Figure 3A*). For example, individuals with 9q iciHHV-6B have a character-istic (CTATGG-TTAGTG-CTATGG) motif (pink-cyan-pink repeats) in the central region of DR$_R$-pvT1, as well as a rare (TTAGAG) repeat (peach) that is found infrequently outside this group. Also, viral sequences assigned to the 17p  (major) iciHHV-6B group by genome sequence identity and shared 17p  subtelomere-iciHHV-6B junction sequences (HGDP01065, DER512, 801018, and ORCA1622) (*Zhang et al., 2017*) had similar DR$_R$-pvT1 repeat patterns with a characteristic (GTAGTG) (blue) repeat at the fifth position, replacing the (TTAGTG) (cyan) repeat seen in almost all other pvT1 repeat patterns (*Figure 3A*, *Figure 1—figure supplement 1*, *Supplementary file 3*).

From these observations, we hypothesized that DR$_R$-pvT1 repeat patterns may be used to predict both the integration site for newly identified iciHHV-6B carriers and the identity of closely related viral strains, without the need to sequence viral genomes. Based on similarity between DR$_R$-pvT1 repeat patterns, the integration site was predicted for nine iciHHV-6B genomes for which genome sequences were not available (*Figure 3A*). For example, the repeat patterns in YOR546, CRL-1730 and 6-iciHHV-6B were identical or almost identical to those in a large group of iciHHV-6B genomes with a 9q integration site (*Nacheva et al., 2008*; *Shioda et al., 2018*). PCR amplification of the subtelomere-iciHHV-6B junction was used to validate several of the predictions arising from these results. For example, DR$_R$-pvT1 analysis placed 410005 and 308006 in the same phylogenetic clade as the 401027 iciHHV-6B genome (*Figures 2 and 3A*), and common ancestry was confirmed by sequence similarity across the subtelomere-iciHHV-6B junction fragment (amplified by primer 2p2, *Figures 2 and 3B and C*) in all three samples. Similarly, the iciHHV-6B genomes in d44, LAT018, NWA008, and KEN071 share DR$_R$-pvT1 repeat patterns and subtelomere-iciHHV-6B junction fragments similar to the large group of iciHHV-6B genomes integrated at 17p  (17p  major; *Figure 3B and C*, *Zhang et al., 2017*). In summary, the integration sites of iciHHV-6B genomes that had not been sequenced were predicted based on a high degree of similarity between DR$_R$-pvT1 repeat patterns, and six of these predictions were subsequently tested and validated by an independent method.

## Chromosomal integration of acqHHV-6B

Telomeric integration provides a means by which acqHHV-6A/6B may achieve latency. Although de novo integration has been shown to occur in cell culture (*Arbuckle et al., 2010*; *Arbuckle et al., 2013*; *Collin et al., 2020*; *Gilbert-Girard et al., 2020*; *Gravel et al., 2017*; *Wallaschek et al., 2016b*),

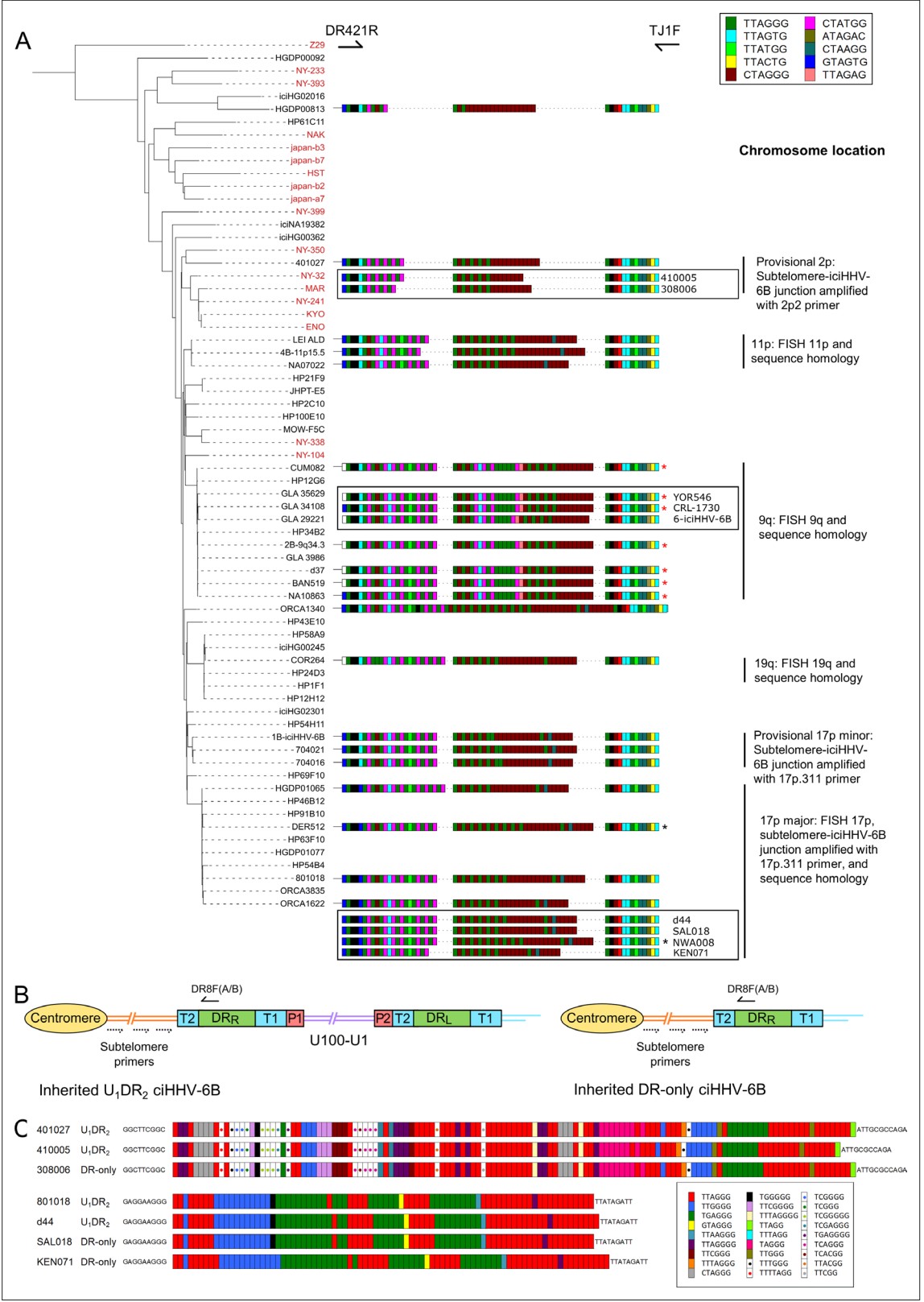

**Figure 3.** pvT1 repeat patterns are similar between strains of iciHHV-6B known to have the same ancestral integration site and those predicted to share that integration site. (**A**) Distance-based phylogenetic tree of selected iciHHV-6B (black names) and acquired HHV-6B (red names) genomes. Where DNA was available, $DR_R$-pvT1 repeat patterns from 19 iciHHV-6B samples in the phylogenetic tree are shown. The repeats in the pvT1 region are color-coded as for **Figure 1** and shown in the key, top right. Black vertical bars on the right indicate groups of iciHHV-6Bs for which the integration

*Figure 3 continued on next page*

*Figure 3 continued*

site has been determined by fluorescent in situ hybridization (FISH), phylogeny, subtelomere-iciHHV-6B junction analysis, or a combination of these. In addition, red asterisks identify samples from the 9q-iciHHV-6B clade that share identical pvT1 repeat patterns. pvT1 repeat patterns are more variable between samples that belong to the 17p (major) iciHHV-6B clade with black asterisks identifying samples that share identical pvT1 patterns. $DR_R$-pvT1 repeat patterns within black boxes identify iciHHV-6B samples for which the integration site has been predicted based on $DR_R$-pvT1 similarity. For some of these samples, the integration site has been validated by subtelomere-iciHHV-6B junction analysis (410005, 308006, d44, SAL018, NWA008, and KEN071). (**B**) Line diagram showing the position of DR8F(A/B) and potential subtelomere primers used to amplify and sequence subtelomere-iciHHV-6B junctions. The subtelomere-iciHHV-6B integration site can be amplified whether the integrated viral genome is full length or iciHHV-6B-DR only. (**C**) Diagram showing the pattern of canonical (TTAGGG) and degenerate telomere-like repeat maps across the subtelomere-HHV-6B junction at two different chromosome ends. The subtelomere-HHV-6B repeat maps from 401027, 410005, and 308006 were generated following amplification with primers 2p2 and DR8F(A/B). They are very similar demonstrating common ancestry (top three repeat maps). The subtelomere-HHV-6B repeat maps from 801018, d44, SAL018, and KEN071 generated by amplification with 17p311 and DR8F(A/B) (bottom four maps) are very similar to the group of iciHHV-6B genomes in the 17p (major) clade some of which have been shown to be integrated in 17p. The 308006, SAL018, and KEN071 samples contain iciHHV-6B DR-only. The key (bottom right) shows the color-coding for the diverse degenerate repeats found across the subtelomere-HHV-6B repeat maps.

there is little evidence that it occurs contemporaneously in vivo (*Moustafa et al., 2017*). To investigate this, we collected saliva DNA samples from 52 healthy donors and used triplicate digital droplet PCR (ddPCR) assays to quantify HHV-6B genome copy number per cell (*Figure 4A*). Using this sensitive approach, low-level HHV-6B was detected in 44 samples (84.6%) with a mean value of 0.00145 copies per cell (range 0.0000233–0.0125). Two kidney DNA samples, K1 and K10, were also positive for HHV-6B at 0.000525 and 0.0124 copies per cell, respectively (*Figure 4A and D*).

Single TElomere Length Analysis (STELA) is a PCR-based method that amplifies full-length telomere molecules from a selected chromosome end, determined by the specificity of the primer located internally to the telomeric repeats (*Baird et al., 2003*). Typically, multiple STELA reactions, each containing low quantities of genomic DNA (usual range 250–1000 pg or 38–152 cell equivalents per PCR), are conducted in parallel and processed by Southern blot hybridization to detect amplified telomere molecules of various lengths. The data collected from each STELA reaction is pooled to estimate average telomere length (*Jeyapalan et al., 2008*). Previously, we used STELA to detect and measure the length of iciHHV-6A/6B-associated telomeres at $DR_L$-T1 (*Huang et al., 2014*). To determine whether some copies of acqHHV-6B are integrated into telomeres, we aimed in this study to detect these potentially rare events using STELA in samples that had low but measurable levels of HHV-6B DNA (*Figure 4A*). Given the low abundance of HHV-6B in these samples, the quantity of saliva DNA added to each STELA PCR was increased (range 2–5 ng or 300–760 cell equivalents per PCR) and up to 900 parallel STELA reactions were processed. In addition, an aliquot of each completed STELA PCR was diluted and subjected to a second round of nested PCR with primers DR421R and TJ1F to amplify $DR_L$-pvT1. Successful amplification in the second round confirmed that an HHV-6B-associated telomere had been amplified in the primary STELA round (*Figure 4B and C*). Sanger sequencing was used to verify the $DR_L$-pvT1 repeat patterns from STELA products matched that amplified in bulk genomic DNA from the donor. Various steps were taken to avoid false-positive results. These include the use of newly designed STELA primers for these experiments (*Figure 4B*) and conducting STELA on genomic DNA from the HT1080 cell line (HHV-6A/6B negative) to check for potential non-specific amplification from another telomere or elsewhere in the human genome. No STELA products were generated from HT1080 among 900 STELA reactions, total 4.5 µg DNA screened. This two-step STELA procedure, conducted on DNA from six saliva and two kidney samples, identified a small number of amplicons from some samples with acqHHV-6B (*Supplementary file 4*). The proportion of HHV-6B genomes from which a telomere was amplified was estimated using the HHV-6B copy number per cell (determined by ddPCR) and by estimating the number of cell equivalents of DNA used per STELA reaction, assuming 6.6 pg DNA per cell (*Figure 4D*, *Supplementary file 4*). For the eight samples analyzed, an average of 0.95% (range 0–1.98%) of HHV-6B genomes resulted in pvT1 amplification, indicative of telomeric integration. The low copy number of HHV-6B in these samples, combined with reduced PCR amplification efficiency of longer molecules (telomeres in this case), has a stochastic effect on the potential to detect an HHV-6B-associated telomere in each STELA reaction. For these reasons, the error rate associated with detection of acqHHV-6B-associated telomere is undetermined and the integration frequencies should be interpreted cautiously.

We then sought to measure the proportion of acqHHV-6B genomes that are *not* integrated in saliva samples, using ddPCR to quantify the copy numbers of DR and PAC1. PAC1 and PAC2 are genome

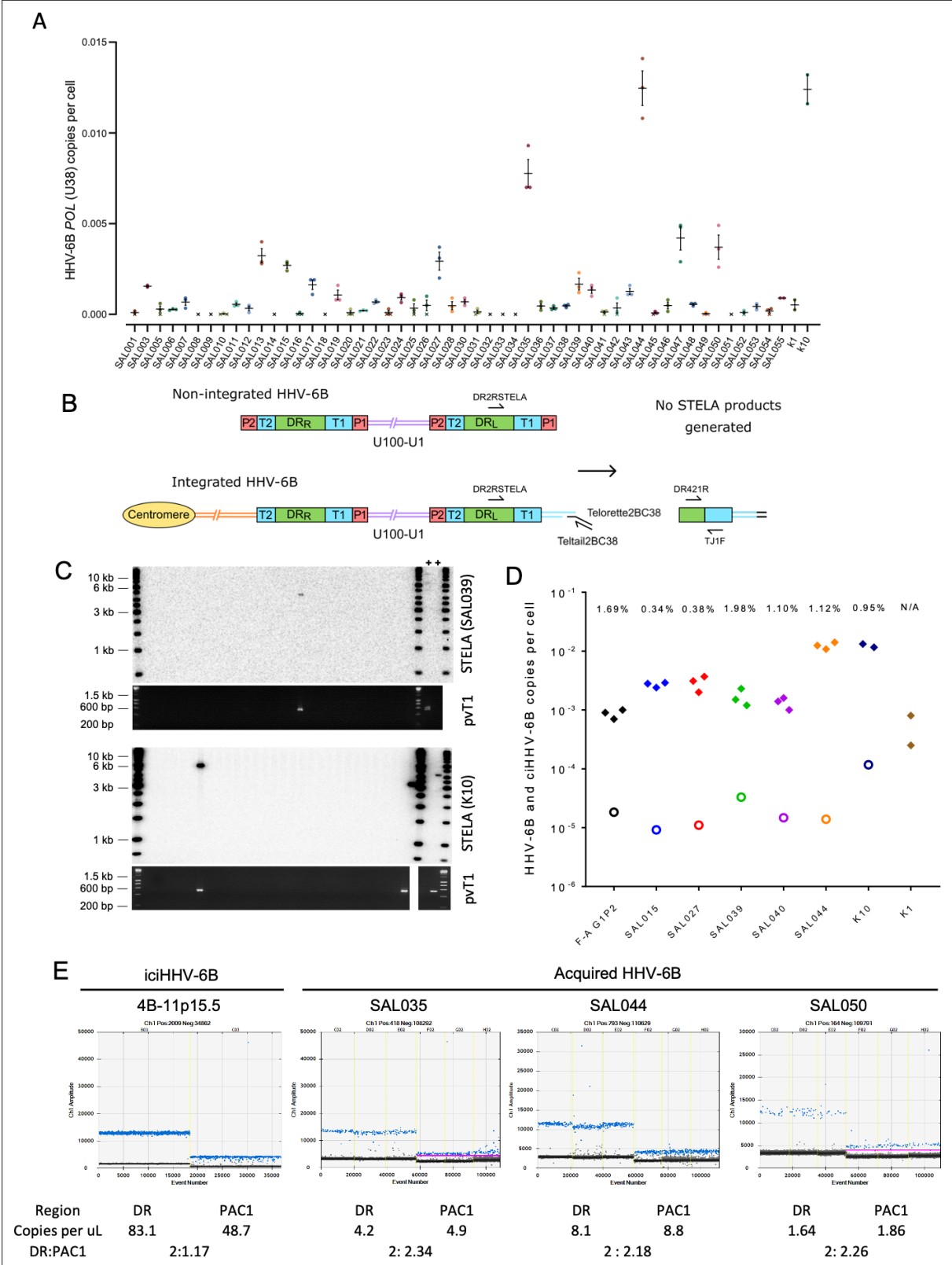

**Figure 4.** Evidence of telomere formation at the distal end of HHV-6B genomes in individuals with community-acquired HHV-6B. (**A**) Graph showing HHV-6B copy number per cell estimated from HHV-6B *POL* (U38) vs. RPP30 duplex digital droplet PCR (ddPCR). Triplicate results are shown for saliva DNA from 52 individuals and kidney samples from 2 individuals. Crosses show where zero copies per cell were estimated for a single replicate, and black crosses indicate that zero copies per cell were estimated from all three replicates. The mean and standard error are shown. (**B**) Single TElomere

*Figure 4 continued on next page*

*Figure 4 continued*

Length Analysis (STELA) primers, DR2RSTELA and Telorette2BC38/TeltailBC38, that were used to amplify HHV-6B-associated telomeres are shown on an integrated HHV-6B genome. DR421R and TJ1F primers were used to amplify DR$_L$-pvT1 in secondary nested PCRs. In non-integrated HHV-6B, STELA products cannot be generated as there is no HHV-6B-associated telomere. The copies of PAC1 (P1) and PAC2 (P2) are shown. (**C**) Examples of HHV-6 STELA from two DNA samples (SAL039 and K10) with acqHHV-6B. Upper panels show the outcome of 30 STELA reactions followed by agarose gel electrophoresis and Southern blot hybridization to radiolabeled (TTAGGG)$_n$ telomere-repeat probe. In a small number of reactions, an amplified HHV-6B-associated telomere band was detected. An aliquot of the products from each primary STELA PCR was diluted and used as input for a secondary, nested PCR to amplify DR$_L$-pvT1. As shown in the agarose gel photograph below, DR$_L$-pvT1-positive reactions corresponded to the STELA reactions that generated HHV-6B-associated telomere band. The 6-iciHHV-6B DNA was used as a positive control (+) for STELA and DR$_L$-pvT1 secondary PCR. (**D**) Graph showing the total copy number of HHV-6B per cell measured by ddPCR (filled diamonds); the estimated number of copies of integrated HHV-6B per cell from STELA and pvT1 PCR (open circles). (**E**) Measuring the DR:PAC1 ratio by ddPCR. 1D EvaGreen ddPCR plots of one iciHHV-6B (4B-11p15.5) cell line and acqHHV-6B in three saliva samples (SAL035, SAL044, and SAL050). Triplicate reactions for a DR amplicon (DR6B-F and DR6B-R primers) and a PAC1 amplicon (PAC1F and PAC1R-33 primers) are shown for each sample. The absolute copy number of each amplicon per µL ddPCR reaction and the ratio between DR and PAC1 amplicons are shown below each plot.

packaging signals located at the end of T1 and T2, respectively (*Figure 4B*). The linear, unintegrated genome has two copies of DR and two copies of PAC1 and PAC2, whereas the integrated genome has two copies of DR but retains only a single copy of PAC1 and PAC2 (*Figure 4B*; *Arbuckle et al., 2010*; *Huang et al., 2014*). As expected, the ratio of DR:PAC1 in an iciHHV-6B sample (4B-11p15.5) was approximately 2:1, whereas the ratio in acqHHV6-B samples in saliva samples was approximately 2:2 (*Figure 4E*). This shows that most acqHHV-6B genomes in saliva samples are not integrated, consistent with the STELA data that revealed only low-level telomeric integration of acqHHV-6B genomes in saliva DNA samples.

## Partial excision of iciHHV-6B genomes and novel telomere formation at DR$_L$-T2

We showed previously that iciHHV-6A/6B genomes in lymphoblastoid cell lines (LCLs) can be truncated at DR$_L$-T2, resulting in loss of the terminal DR$_L$ and formation of a novel short telomere at the breakpoint (*Huang et al., 2014*; *Wood and Royle, 2017*). We proposed that truncation is achieved through a t-loop-driven mechanism in which the 3′ G-rich overhang at the end of the telomere strand invades DR$_L$-T2, facilitating excision of the terminal DR$_L$ as a DR-containing t-circle and leaving a novel telomere at the excision point with a length limited by the length of T2 for the particular iciHHV-6A/6B genome. For the iciHHV-6B samples used in this study, T2 ranged in length from 16 to 31 telomere repeats (96–186 bp). Here, we measured the frequency of iciHHV-6B truncations at DR$_L$-T2 in seven LCLs, nine blood DNAs (from white blood cells), two pluripotent cell lines, three sperm DNA and two saliva DNA samples (*Figure 5A-C*, *Supplementary file 5*). The average frequency of DR$_L$-T2 truncations with novel telomere formation was 1 in 120 cells, but there were significant differences between DNAs from different cell types. The number of truncations per cell was lowest in blood and saliva DNA, with an average of 1 in 300 (0.0033 per cell) and 1 in 320 (0.0031), respectively, and approximately double this in sperm at 1 in 160 cells (0.00627 per cell). An astonishingly high number of DR$_L$-T2 truncations were observed in LCLs (0.015 per cell) and pluripotent cells (0.014 per cell), equating to approximately 1 in every 70 cells. Furthermore, released circular DR molecules were detected using an inverse PCR approach (*Figure 5—figure supplement 1A*).

The histone deacetylase inhibitor, trichostatin-A (TSA), has been shown to promote iciHHV-6A reactivation in cell lines and cultured T cells from iciHHV-6A/6B individuals (*Arbuckle et al., 2010*; *Arbuckle et al., 2013*), and also to increase the abundance of t-circles (*Zhang et al., 2019*). To explore whether the frequency of truncations with short telomere formation at DR$_L$-T2 could be influenced by the chromatin state of the iciHHV-6B genome, the 4B-11p15.5 LCL was treated with TSA. The cells were grown in medium supplemented with various concentrations of TSA for approximately 5 days, and the DR$_L$-T2 truncation assay (DR$_L$-T2-STELA) was conducted on DNA extracted from TSA-treated and control cells (*Figure 5D*). The frequency of truncation events at DR$_L$-T2 increased significantly from 0.0162 to 0.0290 per cell at the highest TSA concentration. This suggests that iciHHV-6B chromatin conformation influences the chance of t-loop formation at the telomere-like repeats within the viral genome and subsequent excision events.

As stated above, loss of the terminal DR$_L$ via t-loop formation and excision at DR$_L$-T2 was expected to generate a short novel telomere detected as a STELA fragment consisting of a flanking region of DR$_L$

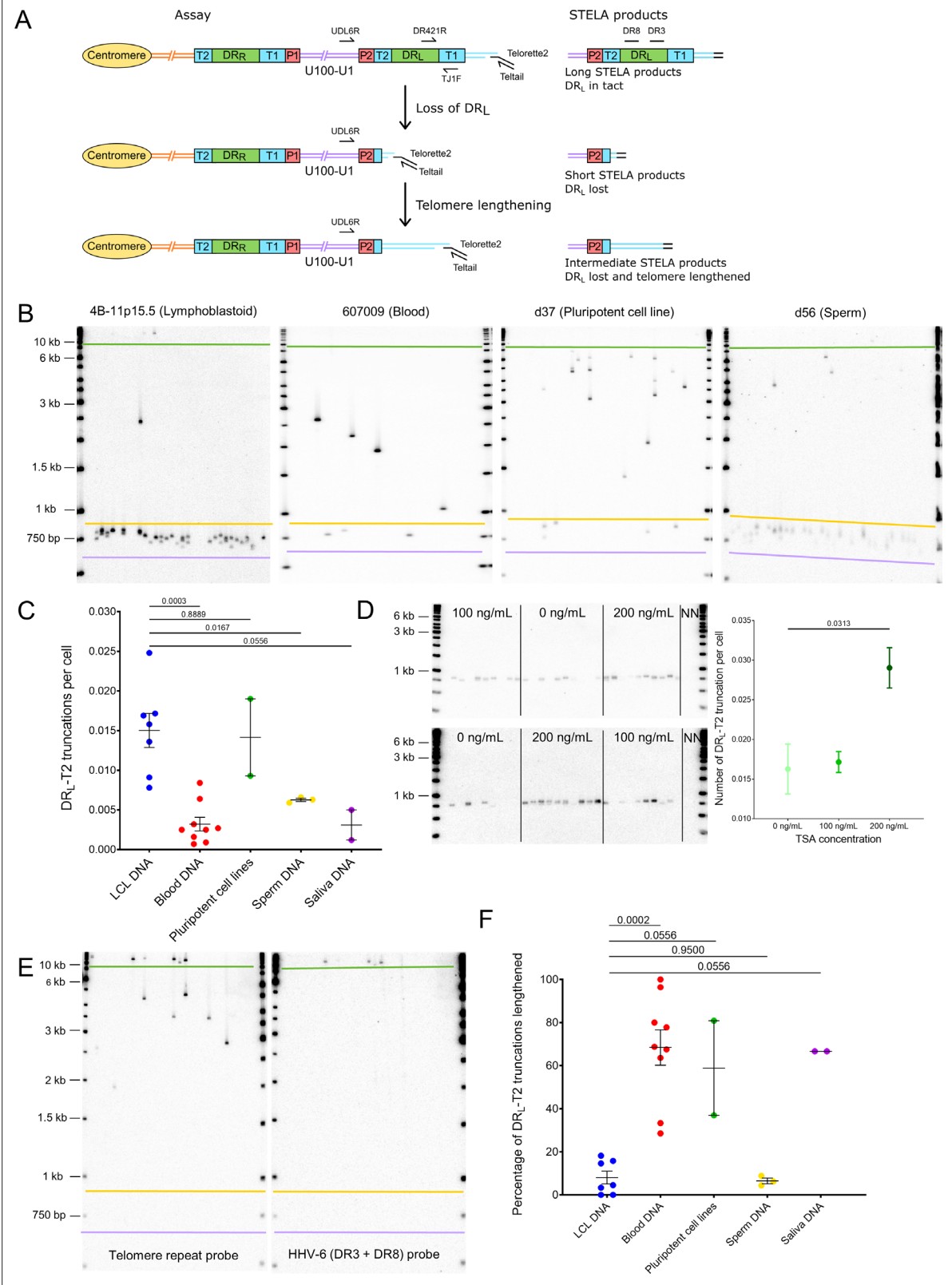

**Figure 5.** Quantifying iciHHV-6B truncations at DR$_L$-T2 and assessment of the newly formed telomere length. (**A**) Diagram showing the loss of DR$_L$ from iciHHV-6B, formation of a novel telomere at DR$_L$-T2, and the potential lengthening of the newly formed telomere by a telomere maintenance mechanism. Single TElomere Length Analysis (STELA) primers (Teltail and Telorette 2) and the primer UDL6R, in the unique region of the iciHHV-6B genome, were used to amplify HHV-6B-associated telomeres. Long STELA products (>8.6 kb) are generated from intact copies of the iciHHV-6B

*Figure 5 continued on next page*

*Figure 5 continued*

genome and include amplification through DR$_L$ to the end of the telomere. Short STELA products (0.7–0.9 kb) are generated when DR$_L$ has been lost and a novel telomere has formed at DR$_L$-T2. Intermediate sized STELA products are generated when the novel telomere formed at DRL-T2 has been lengthened. All three types of STELA amplicons contain TTAGGG telomere repeats and hybridize to a telomere repeat probe. The relative positions of the DR3 and DR8 probes used to detect DR sequences are shown. (**B**) Representative Southern blots exhibiting amplicons generated by HHV-6B STELA with the UDL6R flanking and Telorette 2 primers. Each blot displays products generated from 24 to 32 parallel STELA PCRs in different cell types from iciHHV-6B carriers. These include a lymphoblastoid cell line (LCL) from individual 4B-11p15.5; white blood cells from individual 607009; a pluripotent cell line d37; sperm DNA d56. Amplicons detected between the mauve and yellow lines are short STELA products (expected size range 0.7–0.9 kb). The bands detected between the yellow and green lines are intermediate length STELA products, representative of newly formed telomeres at DR$_L$-T2 that have been lengthened. Long STELA amplicons, above the green line, are products generated from full-length copies of iciHHV-6B and include DR$_L$. (**C**) Graph showing the estimated number of DR$_L$-T2 truncations per cell, including those that have been lengthened, in DNA from different iciHHV-6B individuals and cell types. Means with standard error and the p-values from Mann–Whitney ranked sum test to compare truncation frequencies (LCL vs. other cell types) are shown. (**D**) Representative UDL6R-STELA blots from the 4B-11p15.5 cell line treated with 0, 100, and 200 ng/mL of trichostatin-A (10 parallel STELA PCRs per condition) and a graph showing the frequency of truncations per cell (mean and standard error). Data were derived from UDL6R-STELA conducted twice (technical replicates) using DNA extracted from treated and untreated cells grown in triplicate (biological replicates). p-values were from a Wilcoxon test. (**E**) DR$_L$-T2 truncation assay on blood DNA from the 401027 iciHHV-6B carrier demonstrating that the long STELA products (above the green line) contain DR sequences that hybridize to the telomere repeat probe (left panel) and the combined DR3/DR8 probe (right panel). Whereas the intermediate length amplicons (between yellow and green lines) only hybridize to the telomere repeat probe (left panel) demonstrating that each amplicon represents newly formed telomere at DR$_L$-T2 that has been lengthened. The two Southern blots display products detected across 24 parallel STELA PCRS. No short amplicons (0.7–0.9 kb) were detected in this replicate of the assay for this sample. (**F**) The graph shows the percentage of newly formed telomeres at DR$_L$-T2 that were lengthened in samples from various unrelated iciHHV-6B carriers and cell types. Means with standard error and the p-values from Mann–Whitney ranked sum test to compare truncation frequencies (LCL vs. other cell types) are shown.

The online version of this article includes the following figure supplement(s) for figure 5:

**Figure supplement 1.** Detection of excised circular DR-only molecules, sequencing of a lengthened new telomere at DR$_L$-T2, and detection of telomerase activity.

of approximately 600 bp plus a telomere repeat array limited to the length of the DR$_L$-T2 (*Figure 5A*). Some of the DR$_L$-T2-STELA amplicons were longer than the expected 0.7–0.9 kb (*Figure 5B*). We showed that these amplicons lack DR3 or DR8 sequences (*Figure 5E*), thus indicating that they are not molecules truncated at different sites within DR$_L$. We hypothesized that the intermediate length amplicons (*Figure 5B*) could have arisen by telomerase-mediated lengthening of newly formed telomeres at the DR$_L$-T2 truncation site. To address this, six of the lengthened amplicons from three DNA samples (4B-11p15.5, d56, and NWA008) were sequenced. All comprised the expected flanking sequence followed by a uniform array of (TTAGGG) repeats exceeding the known length of DR$_L$-T2 for the sample, thus showing that some of the novel telomeres formed at DR$_L$-T2 had been lengthened (*Figure 5—figure supplement 1B*). Telomerase activity was detected at a low level in the 4B-11p15.5 and other iciHHV-6B LCLs (*Figure 5—figure supplement 1C*), consistent with a telomerase-mediated lengthening of newly formed telomeres.

The percentage of DR$_L$-T2 telomeres that had undergone lengthening varied between the iciHHV-6B samples and cell types (*Figure 5F*). The proportion of lengthened telomeres was higher in the pluripotent cell lines, blood DNA, and saliva samples, and lower in the LCLs and sperm samples (*Supplementary file 5*). This may reflect the level of telomerase activity in the cells or their progenitors, but it may also indicate variable responses to extremely short telomeres.

## Partial excision of iciHHV-6B genomes and novel telomere formation at DR$_R$-T1

Processes driven by t-loop formation can also facilitate the release of U in its entirety with a single DR from iciHHV-6B (reconstituted with both PAC1 and PAC2) as a circular (U-DR) molecule (*Arbuckle et al., 2013*; *Borenstein and Frenkel, 2009*; *Huang et al., 2014*; *Wood and Royle, 2017*). The reciprocal product of U-DR excision is expected to be a truncation of the iciHHV-6B genome at DR$_R$-T1 with new telomere formation. To explore this, we exploited the variable nature of pvT1. Differences between DR$_R$-pvT1 and DR$_L$-pvT1 within a single iciHHV-6B genome were found in many of the iciHHV-6B samples analyzed (24/35, 68.6%; *Figure 6A*, *Figure 6—figure supplement 1*). These differences were usually a consequence of loss or gain of a small number of repeats, although in some cases they involved single-base substitutions that converted one repeat type to another. These differences

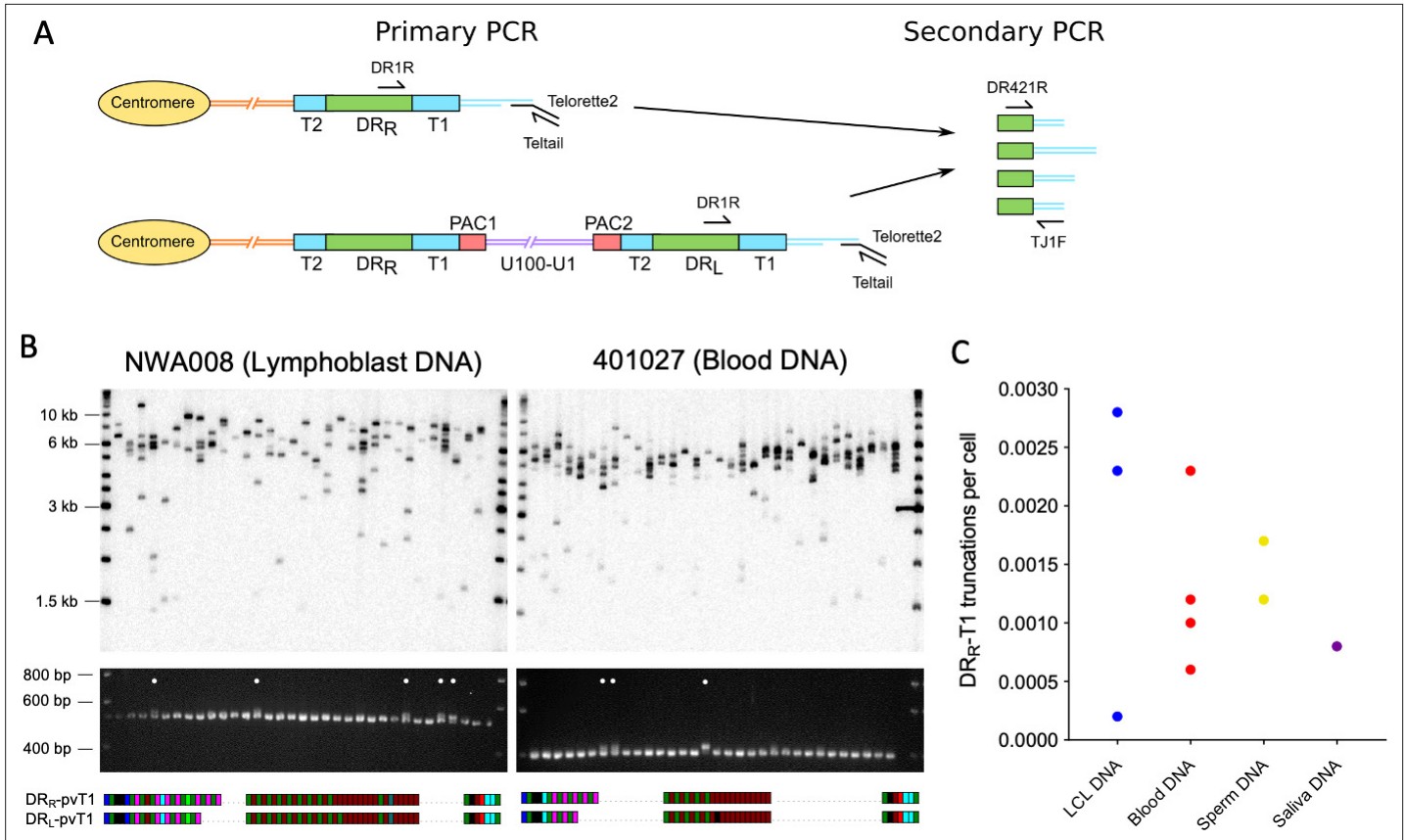

**Figure 6.** Measuring the frequency of iciHHV-6B truncations at $DR_R$-T1 associated with novel telomere formation. (**A**) Schematic of the two-step assay used to differentiate between iciHHV-6B-associated telomeres at $DR_R$-T1 and $DR_L$-T1. Telomeres at $DR_R$-T1 and $DR_L$-T1 were PCR amplified using the Single TElomere Length Analysis (STELA) primers, DR1R with Telorette 2/Teltail. Subsequently, secondary nested PCRs were used to amplify pvT1 (primers DR421 and TJ1F) followed by size separation to distinguish pvT1 from $DR_R$-T1 or $DR_L$-T1. (**B**) Detection of telomeres at $DR_R$-T1 or $DR_L$-T1 in two iciHHV-6B samples (NWA008 and 401027) with known length differences between pvT1 in the DRs. Top images show telomere amplicons derived from $DR_L$ and $DR_R$ in the iciHHV-6B samples. The telomeres were amplified in 32 parallel STELA PCRs using the DR1R flanking primer and detected by Southern blot hybridization to a radiolabeled $(TTAGGG)_n$ probe. Lower images show ethidium bromide-stained agarose gels of size-separated pvT1 sequences amplified from the corresponding STELA reaction in track shown in the panel above. White dots identify the PCR reactions that contain pvT1 from $DR_R$-T1. Below each set of panels are the pvT1 repeat patterns from $DR_R$-T1 or $DR_L$-T1 for the corresponding sample, highlighting the differences. (**C**) Graph showing the frequency of truncations and new telomere formation at $DR_R$-T1 in iciHHV-6B samples from unrelated individuals and cell types. The number of truncations per cell was calculated using the number of reactions from which $DR_R$-pvT1 was amplified divided by the cell equivalents estimated from the total quantity of DNA screened (6.6 pg or 3.3 pg DNA per cell was assumed for diploid and haploid cells, respectively).

The online version of this article includes the following figure supplement(s) for figure 6:

**Figure supplement 1.** The majority of iciHHV-6B genomes show sequence differences between $DR_R$-pvT1 and $DR_L$-pvT1.

made it possible to measure the frequency of truncations and new telomere formation at $DR_R$-T1 via a two-step process.

First, the iciHHV-6B-associated telomeres were amplified using STELA with low input of genomic DNA per parallel reaction (500 pg/reaction) and in the presence of the DR1R primer that can generate products from a telomere at $DR_L$-T1 (from normal, full-length iciHHV-6B) or $DR_R$-T1 (following U-DR excision). Second, to distinguish between these STELA products, the PCR amplicons were subjected to nested PCR to amplify $DR_L$-pvT1 and $DR_R$-pvT1 (**Figure 6A**). The sizes of the pvT1 amplicons indicated whether the STELA products in the primary PCR had been derived from telomeres at $DR_L$ or $DR_R$, and selected sequencing was used to confirm they contained the expected $DR_R$-pvT1 pattern of repeats (**Figure 6B**). The size difference between the $DR_L$-pvT1 and $DR_R$-pvT1 amplicons is usually small, influencing the ability to size-separate the amplicons. Consequently, 10 iciHHV-6B samples, with larger pvT1 size differences, were selected and truncations at $DR_R$-T1 were detected in all. In summary,

the frequency of newly formed telomeres at $DR_R$-T1 varied between samples (0.0002–0.0029 per cell; *Figure 6C*, *Supplementary file 6*).

## Generation of DR-only iciHHV-6B by partial excision of full-length iciHHV6-B genomes in the germline

The frequency of truncations and new telomere formation at $DR_R$-T1 in sperm DNA from iciHHV-6B carriers raised the strong possibility that such events could explain how individuals may inherit integrations consisting only of DR (DR-only iciHHV-6A/6B) (*Huang et al., 2014*; *Liu et al., 2020*; *Ohye et al., 2014*). To date, we have identified three individuals who carry iciHHV-6B DR-only (308006 and SAL018 in the present study, and KEN071 *Huang et al., 2014*). Comparison of pvT1 repeat patterns suggested shared ancestry between these individuals with DR-only iciHHV-6B and those with full-length iciHHV-6B (*Figure 3A*). To investigate this, attempts were made to amplify and sequence the subtelomere-iciHHV-6B junction, based on previously described junction fragments (*Tweedy et al., 2016*; *Zhang et al., 2017*) and using primers that anneal in the subtelomeric regions of various chromosomes (*Figure 3B*). Using this approach, we found that the DR-only iciHHV-6B in SAL018 and KEN071 share the common 17p (major) subtelomere-iciHHV-6B junction that is also found in 801018, d44, and others (*Figure 3C*). Similarly, the pattern of telomere and degenerate repeats across a subtelomere-iciHHV-6B junction (amplified by subtelomere primer 2p2) in the DR-only iciHHV-6B sample 308006 closely matched those in full-length iciHHV-6B individuals 401027 and 410005 (*Figure 3C*). The existence of shared integration sites for the full-length and DR-only iciHHV-6B carriers at two different chromosome ends clearly establishes that the DR-only status has arisen independently, on at least two occasions, by loss of U and one copy of DR in the germline of an ancestor with a full-length iciHHV-6B genome.

## Evidence of viral transmission from iciHHV-6B carrier mother to non-iciHHV-6B son

It has been shown that iciHHV-6A can reactivate in an immune-compromised setting (*Endo et al., 2014*) and there is evidence of transplacental transmission of reactivated iciHHV-6A/6B (*Gravel et al., 2013*). Despite these examples, the high prevalence of HHV-6B in populations has made it difficult to determine how often iciHHV-6B reactivation occurs. The evidence for frequent partial or complete release of iciHHV-6B genomes in somatic cells and the germline presented above suggests that opportunities for reactivation may be more common than currently appreciated. To explore this, we used $DR_R$-pvT1 analysis to investigate the relationship between strains of iciHHV-6B and low-level acqHHV-6B within families. In one family (Rx-F6a), the mother (G3P1) had a low acqHHV-6B load in saliva (0.00035 copies per cell), but both her children were iciHHV-6B carriers. The $DR_R$-pvT1 repeat patterns in the children were identical and presumed to have been inherited from the father, who was not available for testing. Clearly the mother in this family had a different HHV-6B strain with a distinct $DR_R$-pvT1 repeat pattern (*Figure 7A*, *Figure 7—figure supplement 1*).

Quantification of HHV-6B in saliva samples from another family (Rx-F3a, *Figure 7B*) showed that the mother (G3P2) was an iciHHV-6B carrier and that her daughter (G4P3) had approximately two copies of iciHHV-6B per cell. Identical $DR_R$-pvT1 repeat patterns between the mother and daughter proved maternal inheritance of one iciHHV-6B copy. A second, distinct $DR_R$-pvT1 repeat pattern in the daughter was assumed to represent iciHHV-6B inherited from the father, who was not available for testing. The son (G4P1) had 0.00025 copies of HHV-6B per cell in saliva DNA, consistent with the level expected from acqHHV-6B infection (*Figure 7—figure supplement 1*). Importantly, the $DR_R$-pvT1 repeat pattern in the son (G4P1) was identical to that in the mother and one copy of iciHHV-6B in his sister. This repeat pattern was not seen outside this family among 102 $DR_R$-pvT1 repeat patterns (*Figure 1—figure supplement 1*). These observations strongly suggest that iciHHV-6B genome excision occurred in the mother and that the reactivated HHV-6B was then transmitted to the non-iciHHV-6B son, who retained residual viral sequences in his saliva.

## Discussion

The discovery of germline transmission of chromosomally integrated HHV-6A/6B genomes has raised questions about the possible relationship between iciHHV-6A/6B carrier status and lifelong disease

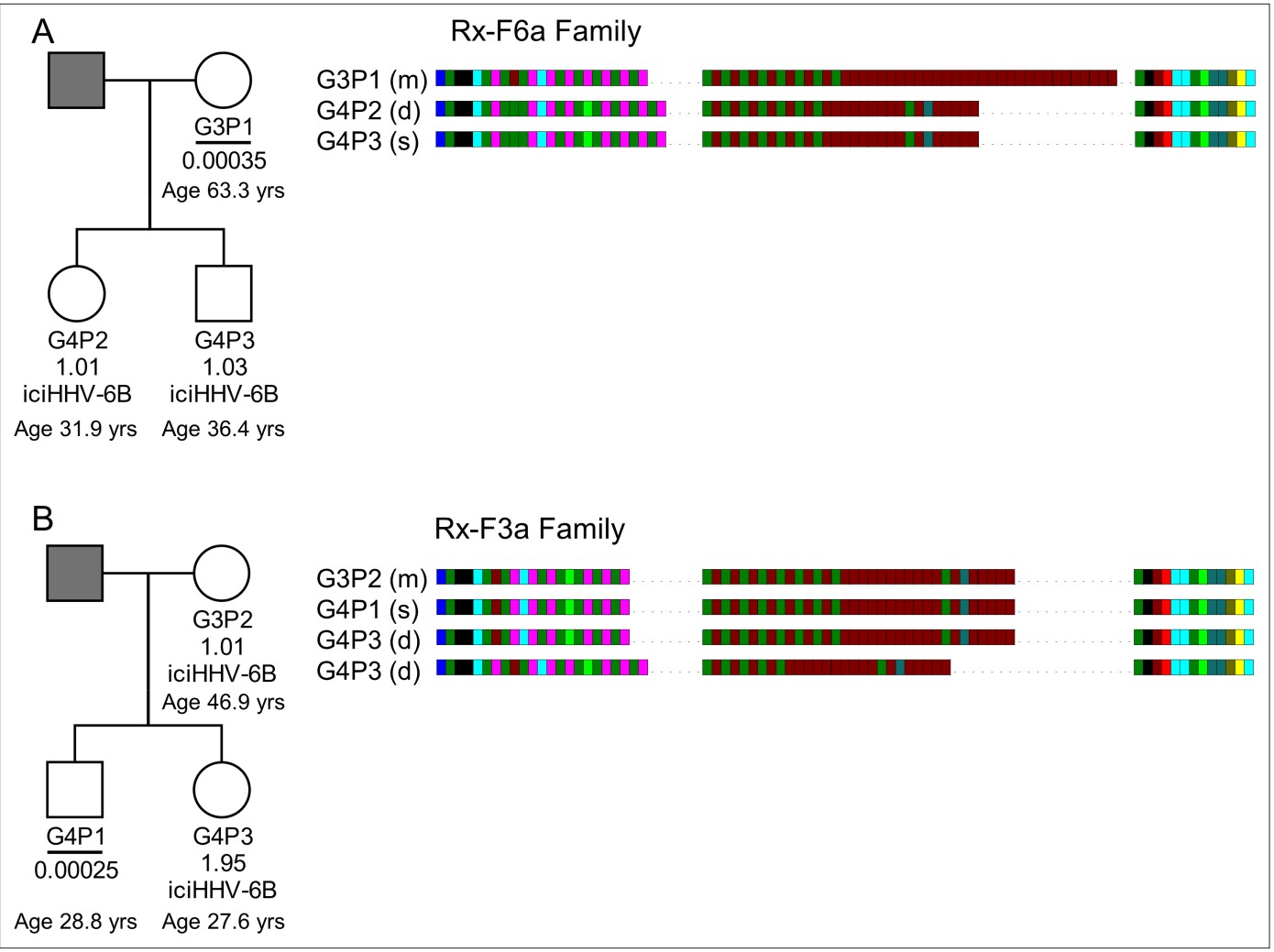

**Figure 7.** HHV-6B copy number and DR$_R$-pvT1-repeat patterns can be used to identify potential iciHHV-6B reactivation and transmission within families. (**A**) In family Rx-F6a, both children are iciHHV-6B carriers with approximately one copy per cell. They share the same DR$_R$-pvT1 repeat map, presumably inherited from the father (gray filled square, not available for testing). The mother has a low level of acqHHV-6B in her saliva (0.00035 copies per cell) and the DR$_R$-pvT1 repeat map is different from her children. (**B**) Evidence of iciHHV-6B reactivation in family Rx-F3a and transmission to non-iciHHV-6B son. Left shows the Rx-F3a family tree with HHV-6 copy number per cell in saliva DNA and iciHHV-6B carrier status. Father (gray filled square) was not available for testing. Right: the DR$_R$-pvT1 repeat patterns from family members. Daughter (G4P3) has two copies of iciHHV-6B, one copy shares the same DR$_R$-pvT1 as seen in her mother (G3P2) and a second copy has a different DR$_R$-pvT1 repeat pattern assumed to have been inherited from her father. The son (G4P1) has a very low level of HHV-6B in his saliva with the same pvT1 repeat pattern as the maternal iciHHV-6B genome. The DR$_R$-pvT1 repeat patterns are also labeled with m (present in mother), d (present in daughter), and s (present in son). The age of tested family members is stated.

The online version of this article includes the following figure supplement(s) for figure 7:

**Figure supplement 1.** Quantification of HHV-6B using POL (U38) digital droplet PCR (ddPCR) in two families (Rx-3FA and Rx-6Fa).

risk (**Gravel et al., 2015**; **Gaccioli et al., 2020**; **Hill et al., 2017**). Events that could have deleterious consequences include full viral reactivation (**Endo et al., 2014**; **Gravel et al., 2013**; **Hall et al., 2010**), intermittent expression of iciHHV-6A/6B genes that may elicit intermittent immune and inflammatory responses over a lifetime (**Peddu et al., 2019**), and an impact on telomere stability and function (**Huang et al., 2014**; **Prusty et al., 2013**; **Wood and Royle, 2017**; **Zhang et al., 2017**).

Progress towards understanding the potential impact of iciHHV-6A/6B carrier status has been advanced by the recent increase in the number of HHV-6A/6B genome sequences available (**Aswad et al., 2021**; **Greninger et al., 2018**; **Zhang et al., 2017**). This has shown that most iciHHV-6A/6B genomes in populations are derived from a small number of ancient telomere-integration events. Here we have expanded the number of sequenced iciHHV-6A/6B genomes from individuals in Europe

and North America and developed the web-based HHV-6 Explorer that can be used to interrogate diversity and functionality between inherited and acquired HHV-6A or HHV-6B genomes. The HHV-6 Explorer includes the sequence of 84 complete or nearly complete iciHHV-6A/6B genomes representative of the various known clades in the phylogenetic network and genome sequences from 28 HHV-6A/6B viruses circulating in populations. Using the HHV-6 Explorer, we displayed two mutations that introduced stop codons in U79 in three of seven iciHHV-6A samples in the 17p clade and in U14 in one of three iciHHV-6B samples in the 17p (minor) clade (*Figure 2—figure supplement 1*), demonstrating that each mutation arose after integration. The presence of an inactivating iciHHV-6A/6B mutation in some but not all descendants of a particular ancestral integration event adds a further complication to understanding the potential lifetime risk for individual iciHHV-6A/6B carriers.

The relationship between iciHHV-6A/6B carrier status and potential associated disease risk may also be influenced by interactions between the integrated viral genome and genes or chromatin on the chromosome carrying the viral genome (*Gravel et al., 2015*; *Wood and Royle, 2017*). It is therefore important to have a rapid method to distinguish between viral strains and integration sites, which is offered by the hypervariable pvT1 region in HHV-6B. pvT1 is a complex repeat region comprising a variety in degenerate repeats, variation in repeat number, and the interspersed nature of different repeats. Nevertheless, pvT1 length variation is mostly attributable to the diverse interspersion of (CTAGGG) and canonical (TTAGGG) telomere repeat sequences in the central region of pvT1 amplicons. Notably, short blocks of (CTAGGG)$_n$ repeats are highly unstable in human telomeres in somatic and germline cells, probably due to replication errors and in vitro studies showed that (CTAGGG)$_n$ repeats bind efficiently to POT1, a component of Shelterin (*Barrientos et al., 2008*; *Baumann and Cech, 2001*; *de Lange, 2018*; *Lim et al., 2009*; *Loayza et al., 2004*; *Mendez-Bermudez et al., 2009*). Furthermore, pvT1 in iciHHV-6B and acqHHV-6B will mutate at different rates and have interconnected but different evolutionary histories, influenced by viral genome integration and excision rates that are currently unknown. Consequently, comparative analysis of the $DR_R$-pvT1 repeat patterns was achieved manually by dividing the pattern of repeats into the proximal, central, and distal regions with respect to the DR421 primer. Where possible, distinctive motifs that distinguish between the pvT1 repeat patterns of various ciHHV-6B clades were identified (*Supplementary file 3*) and these were informed by known relationships between iciHHV-6B genomes based on whole genome sequence similarity, subtelomere-iciHHV-6B junctions, and FISH. Furthermore, the presence of a particular motif in $DR_R$-pvT1 and $DR_L$-pvT1 in a single iciHHV-6B genome indicated that the pattern was present at integration, and this was a useful aid to defining the pvT1 repeat pattern for the clade. As pvT1 repeat patterns from additional samples with known phylogenetic relationships become available, an automated analysis of pvT1 is desirable. Nevertheless, from the data presented it is clear that $DR_R$-pvT1 repeat patterns can be used in many cases as a rapid and inexpensive way to predict the phylogenetic clade and integration site of an iciHHV-6B genome.

$DR_R$-pvT1 analysis is also an excellent tool for tracking acqHHV-6B transmission in families as shown by the evidence that community-acquired HHV-6B transmission usually occurs between family members, with maternal transmission most common. We showed that $DR_R$-pvT1 repeat patterns can be used effectively to discriminate between HHV-6B strains circulating in communities (61/63 different $DR_R$-pvT1 sequences in saliva from healthy non-iciHHV-6B donors in the UK). In the future, $DR_R$-pvT1 analysis could be used to trace patterns of transmission more generally. This may be particularly important in the setting of organ and tissue transplants (*Hill, 2019*). For example, reactivation of iciHHV-6B from a donor tissue could be monitored and differentiated from HHV-6B acquired by the recipient in early childhood or to identify cases of multiple infections by different strains of HHV-6B.

The iciHHV-6B and free HHV-6B viruses in communities must have evolutionary histories that are interlaced, but these are difficult to disentangle as there is little understanding of HHV-6B telomeric integration and iciHHV-6B excision and reactivation. The current picture indicates a modest number of ancient iciHHV-6B clades (in Europe and North America at least) some of which are accumulating mutations that could prevent full reactivation (discussed above). There is also at least one example of an iciHHV-6B genome with high-sequence homology to a group of genome sequences from acqH-HV-6B (*Aswad et al., 2021*; *Greninger et al., 2018*), which complicates interpretation of HHV-6B phylogenetic relationships (*Forni et al., 2020*). Indirectly this suggests that some modern circulating strains retain the capacity for germline integration into telomeres and inheritance or that iciHHV-6B can reactivate and be transmitted, re-entering the reservoir of circulating strains. We demonstrated

that some copies of HHV-6B in saliva acquire a telomere, indicative of integration in somatic cells in vivo and as previously detected in vitro (*Arbuckle et al., 2010*). The viral load in the saliva samples was low, as seen in some other studies (*Leibovitch et al., 2014*; *Leibovitch et al., 2019*; *Turriziani et al., 2014*), and the number of HHV-6B genomes with a telomere was small (*Figure 4*, *Supplementary file 4*), which is consistent with the majority of HHV-6B genomes present as viral particles in saliva (*Jarrett et al., 1990*). The approach we used to detect telomeric integration in saliva could, in principle, be used to detect HHV-6B integration events in sperm from healthy non-iciHHV-6B men (*Neofytou et al., 2009*; *Godet et al., 2015*; *Kaspersen et al., 2012*). This would address the outstanding question of whether current HHV-6B strains circulating in communities can integrate in the germline.

Previously we have proposed that iciHHV-6A/6B genomes can be excised from telomeres in a one- or two-step process as by-products of t-loop processing (*Huang et al., 2014*; *Wood and Royle, 2017*). In this study, we measured the frequency of iciHHV-6B excision events, by detection of novel telomeres at $DR_L$-T2 and for the first time at $DR_R$-T1, in unrelated iciHHV-6B carriers and in different tissues and cell types. The frequencies of iciHHV-6B truncations at $DR_L$-T2 (range, 0.014–0.0031 per cell) and $DR_R$-T1 (range, 0.0002–0.0029 per cell) are not directly comparable because they were ascertained by different approaches that have different efficiencies, but both are surprisingly high in vivo (*Figures 5 and 6*). This demonstrates that tissues in iciHHV-6B carriers must be mosaic for different compositions of the iciHHV-6B genome. The telomeres formed at $DR_L$-T2 were expected to be particularly short, limited to the length of the (TTAGGG) repeat array at T2. However, a cell with a small number of very short telomeres will elicit a telomere-mediated DNA damage response and entry into cellular senescence (*Cesare et al., 2013*; d'*d'Adda di Fagagna et al., 2003*; *de Lange, 2018*; *Takai et al., 2003*) so preventing proliferation of a cell with a potentially unstable residual iciHHV-6B genome. It is therefore remarkable that a high percentage of newly formed telomeres at $DR_L$-T2 were lengthened. The frequency of $DR_L$-T2 telomere lengthening was highest in circulating white blood cells and in two pluripotent cell lines (*Figure 5*) consistent with lengthening by telomerase. From these observations and the increase in viral genome excision events in TSA-treated cells, we also propose that iciHHV-6A/6B genomes are more likely to be excised and reactivated in undifferentiated or pluripotent cells. By extrapolation, the potential for iciHHV-6A/6B genome excision, viral gene expression, or full reactivation has the potential to be deleterious during early development (*Miura et al., 2020*).

Finally, we present two lines of evidence that iciHHV-6B genome can be excised and re-enter the reservoir of circulating HHV-6B strains. First, we identified individuals with full-length iciHHV-6B or DR-only genomes that fall into the same clade in the phylogenetic tree (*Figure 3*). These integrated full-length iciHHV-6B or DR-only genomes share similar pvT1 repeat patterns and subterminal-iciHHV-6B junction sequences. This demonstrates that the different iciHHV-6B structures are carried in the same telomere-allele at the particular chromosome end, and it proves that these DR-only carriers have arisen by germline excision of U-DR in an ancestor. The fact that this has occurred independently at least twice suggests that this is a relatively common event, and this is supported by our evidence that new telomere formation at $DR_R$-T1 occurs at a measurable frequency in sperm DNA. Second, we show an example of probable reactivation of iciHHV-6B in a mother with transmission to her son, who carries a very low load of acqHHV-6B with the same distinctive $DR_R$-pvT1 repeat pattern in his saliva. This observation warrants further research to determine how frequently HHV-6B is transmitted from iciHHV-6B parent to their non-iciHHV-6B children, but it requires the use of a hypervariable marker, such as pvT1, that has the power to discriminate between HHV-6B strains.

# Materials and methods

**Key resources table**

| Reagent type (species) or resource | Designation | Source or reference | Identifiers | Additional information |
|---|---|---|---|---|
| Cell line (*Homo sapiens*) | Lymphoblastoid (LCL) | Various, *Supplementary file 1* | Various, see *Supplementary file 1* | |
| Cell line (*Homo sapiens*) | Pluripotent cell line | ATCC | CRL-1730 (RRID:CVCL_2959) | See *Supplementary file 1* |

*Continued on next page*

*Continued*

| Reagent type (species) or resource | Designation | Source or reference | Identifiers | Additional information |
|---|---|---|---|---|
| Cell line (*Homo sapiens*) | Pluripotent cell line | Dr Rai, University of Leicester | d37 | See *Supplementary file 1* |
| Biological sample (*Homo sapiens*) | DNA from saliva | Various, see *Supplementary file 1* | SAL001 to SAL055; TEL-FA(code) to TEL-FL(code); Rx-F6a; RX-F3a | |
| Biological sample (*Homo sapiens*) | DNA from blood | Various, see *Supplementary file 1* | Various, see *Supplementary file 1* | |
| Biological sample (*Homo sapiens*) | DNA from sperm | d32, d44, d56 | Various, see *Supplementary file 1* | |
| Sequence-based reagent | Oligonucleotides and primers | See *Supplementary file 7* | See *Supplementary file 7* | |
| Sequence-based reagent | Hydrolysis probes | See *Supplementary file 7* | See *Supplementary file 7* | |
| Chemical compound, drug | Trichostatin A | Sigma-Aldrich | T8552 | |
| Commercial assay or kit | ddPCR Supermix for probes without dUTP | Bio-Rad Laboratories | 1863023 | |
| Commercial assay or kit | RPP30 primer/HEX labeled probe mix | Bio-Rad Laboratories | 10031243 | |
| Commercial assay or kit | ddPCR EvaGreen Supermix | Bio-Rad Laboratories | 1864033 | |
| Commercial assay or kit | Zymoclean gel DNA recovery kit | Zymo Research, Irvine, CA | D4002 | |
| Software, algorithm | HHV-6 Explorer | This paper; https://www.hhv6explorer.org/ | | https://github.com/colinveal/HHV6-Explorer. |
| Software, algorithm | Prism software (version 9.0) | GraphPad Software | | |

## Saliva collection and DNA extraction

The study was conducted in accordance with the Declaration of Helsinki and with the approval of the University of Leicester's Research Ethics Committee (refs: 10553-njr-genetics; njr-61d3). Saliva samples were donated by individuals and members of families (all 18 years or older) with informed consent and were given anonymous identifiers at the point of collection (*Garrido-Navas et al., 2020*). Approximately 1.5 ml of saliva was collected in a OraGEN saliva collection tube (Genotek, Ottawa, Canada), and DNA was extracted from 500 µL following the manufacturer's instructions.

## Identification of DNA samples and cell lines with iciHHV-6A/6B or acqHHV-6B

The full list of iciHHV-6A/6B-positive DNA samples and cell lines used in this study is given in *Supplementary file 1* with available information on donor ethnicity and iciHHV-6A/6B chromosomal location. To interrogate any relationship between iciHHV-6A/6B carrier status and heart failure, the BIOlogy Study to TAilored Treatment in Chronic Heart Failure (BIOSTAT-CHF) cohort (*Voors et al., 2016*) was screened. The BIOSTAT-CHF study complied with the Declaration of Helsinki and was approved by the relevant ethics committee in each center, while all participants gave their written, informed consent to participate. The 2470 blood DNAs from Europeans in the BIOSTAT-CHF cohort was screened using PCR assays to detect the viral genome (multiplexed PCR to detect HHV-6A/6B DR3 and human MS32 flanking sequence, as a positive control). Positive samples were rescreened with PCR assays to detect DR5 for HHV-6A, DR6 for HHV-6B and U18 to detect the unique region (*Huang et al., 2014*; *Zhang et al., 2017*). This identified 19 iciHHV-6A/6B-positive samples (9 iciHHV-6A, 0.36%, and 10 iciH-HV-6B, 0.40%, one of which was iciHHV-6B DR-only). Five iciHHV-6B carriers were identified among the saliva donors: three with a single copy of iciHHV-6B, one with two copies of iciHHV-6B, and

one with iciHHV-6B DR-only. The saliva samples were also screened for acqHHV-6B using ddPCR, described below. Additionally, two kidney DNA samples (part of the TRANScriptome of renaL humAn TissuE [TRANSLATE] study *Marques et al., 2015*; *Rowland et al., 2019*) were positive for DR3 by PCR assay and were confirmed as acqHHV-6B by ddPCR.

## HHV-6B quantification by ddPCR

HHV-6B copy number was estimated by duplexed HHV-6B POL (U38) vs. RPP30 (human gene, two copies per diploid cell) ddPCR assays as described previously (*Bell et al., 2014*). Hydrolysis probe ddPCRs (20 µL volumes) consisted of 1× ddPCR Supermix for probes without dUTP (Bio-Rad Laboratories), virus-specific primers (HHV-6B POL F and HHV-6B POL R), and FAM-labeled HHV-6B POL (U38) probe (Eurogentec; *Supplementary file 7*) at 300 nM and 200 nM, respectively, 1× RPP30 primer/HEX-labeled probe mix (Bio-Rad Laboratories), 1 µL XhoI digestion mix consisting of 5 U XhoI restriction enzyme and 1× NEB buffer 2.1 (New England Biolabs), and 10 ng genomic DNA for iciH-HV-6B samples or 200 ng DNA for non-iciHHV-6B samples. Droplets were generated using QX200 Droplet Generator (Bio-Rad Laboratories) with 70 µL Droplet Generation Oil for Probes (Bio-Rad Laboratories). Thermocycling was carried out according to the manufacturer's instructions on a Veriti 96-well Thermal Cylcer (Applied Biosystems) as follows: 95 °C, 10 min; 40 cycles 94 °C 30 s, 60 °C 1 min; 98 °C 10 min. QX200 Droplet Reader and QuantaSoft analysis software (Bio-Rad Laboratories) were used to count droplets and measure fluorescence, and to calculate the estimated copy number. DNA from the HT1080, an osteosarcoma-derived established cell line that lacks HHV-6A/B DNA, and DNA from 6-iciHHV-6B, a cell line with one copy of iciHHV-6B, were used as negative and positive controls, respectively, alongside water no-template control reactions, when optimizing the assays. Subsequently, reactions with HT1080 DNA or water were included in each ddPCR run as a negative control to detect non-specific amplification or contamination. Assays to quantify acqHHV-6B in DNA samples from saliva and kidney were carried out in triplicate to increase the accuracy of the copy number estimates. In addition, as very low viral copy numbers were expected in the non-iciHHV-6B samples, the ddPCR reactions were set up in a PCR-clean room where no HHV-6 PCR had been conducted previously.

As required, the copy number and basic organization of the HHV-6B genome in iciHHV-6B samples were established using HHV-6B POL (U38), HHV-6B DR6, and RPP30 (human gene, two copies per diploid cell) hydrolysis-probe ddPCR assays described above and previously (*Bell et al., 2014*). Duplex ddPCRs of HHV-6B POL (U38) with RPP30 or DR6B (FAM-labeled DR6B probe, Eurogentec) with RPP30 determined the copy number of each region and subsequent comparison established the iciHHV-6B genome composition (e.g., $U_1DR_2$ or DR-only) iciHHV-6B.

Absolute quantification of DR and PAC1 was also determined by ddPCR but using the DNA inter-calating dye, EvaGreen. Given the short length of the PAC1 amplicon, the cycle number, extension time, and DNA input level were optimized to maximize the resolution between the fluorescence level of positive and negative droplets. Subsequently, the assays were conducted as follows: EvaGreen ddPCRs (20 µL volumes) consisted of 1× ddPCR EvaGreen Supermix (Bio-Rad Laboratories), virus-specific primers for DR (DR6B-F and DR6B-R) or PAC1 (PAC1F and PAC1R-33) at 300 nM, 1 µL XhoI digestion mix consisting of 5 U XhoI restriction enzyme and 1× NEB buffer 2.1 (New England Biolabs, Ipswich, MA), and 10 ng genomic DNA for iciHHV-6B samples or 100 ng DNA for non-iciHHV-6B samples. Droplets were generated using QX200 Droplet Generator (Bio-Rad Laboratories) with 70 µL Droplet Generation Oil for EvaGreen reactions (Bio-Rad Laboratories). Thermocycling was carried out on a Veriti 96-well Thermal Cycler (Applied Biosystems) with the following conditions: 95 °C, 5 min; 45 cycles of 94 °C 30 s, 60 °C 1 min; 4 °C 5 min; 90 °C 5 min. QX200 Droplet Reader and QuantaSoft analysis software (Bio-Rad Laboratories) were used to count droplets and calculate absolute copy number of each amplicon per µL of each reaction.

## Cell culture and TSA treatment

LCLs with iciHHV-6B were cultured in RPMI 1640 medium as described previously (*Huang et al., 2014*). The CRL-1730 (RRID:CVCL_2959) adherent human vascular endothelial cell line derived from umbilical cord vein (ATCC, *Shioda et al., 2018*) was cultured in F-12K medium (Gibco/ThermoFisher Scientific) supplemented with 10% heat-inactivated fetal bovine serum (Gibco/ThermoFisher Scientific), 0.1 mg/mL heparin (Sigma-Aldrich, Darmstadt, Germany), and 1% endothelial growth supplement

(BD Biosciences). The adherent mesenchymal stem cell line derived by explant culture from umblical cord, d37 (gifted by Dr Sukhvir Rai, University of Leicester), was cultured in human mesenchymal stem cell growth media (MSCBM hMSC Basal Medium, Lonza, Basel, Switzerland) supplemented with mesenchymal stem cell growth supplements (MSCGM hMSC SingleQuot Kit, Lonza).

For TSA treatment, $2 \times 10^6$ cells from the 4B-11p15.5 iciHHV-6B LCL were suspended in RPMI 1640 medium supplemented with 0, 100, or 200 ng/mL TSA (Sigma-Aldrich) in three biological replicates. After 124 hr (5 days), cells were pelleted at 1200 rpm for 8 min, washed twice with phosphate-buffered saline, and snap-frozen on dry ice. DNA was extracted using phenol-chloroform and precipitated using ethanol. The DNA pellet was washed with 80% ethanol, briefly air-dried, and dissolved in nuclease-free water. $DR_L$-T2 STELA (see below) was conducted on extracted DNA using primer UDL6R to detect telomeres at $DR_L$-T2, as described previously (*Huang et al., 2014*). STELA was carried out twice on DNA from each treatment from each of three biological replicates.

## Whole iciHHV-6A/6B genome sequencing

10 iciHHV-6A and 5 iciHHV-6B genomes were sequenced and annotated as described previously (*Huang et al., 2014*; *Zhang et al., 2017*). Pooled, overlapping PCR amplicons (100 ng) were sheared using a Covaris S220 sonicator to an approximate size of 450 bp. Sequencing libraries were prepared using the LTP library preparation kit for Illumina platforms (Kapa Biosystems, Germany). The libraries were then processed for PCR (seven cycles) using the LTP library preparation kit and employing NEBNext multiplex oligos for Illumina index primer pairs set 1 (New England Biolabs). Sequencing was performed using a NextSeq 500/550 mid-output v2.5 300 cycle cartridge (Illumina, San Diego, CA) to produce 6–9 million paired-end 150 base reads per sample. The annotated sequences were deposited in NCBI GenBank under accession numbers: MW049313-MW049327.

## Genome sequence analysis

94 complete or near-complete HHV-6A/6B sequences were downloaded from NCBI GenBank (*Supplementary file 1*) and, together with the 15 newly sequenced genomes, were aligned using MAFFT (v7.407, *Katoh and Standley, 2013*) with default parameters. The alignment was trimmed using trimAI (v1.4.1, *Capella-Gutierrez et al., 2009*) to remove gaps. Phylogenetic networks were inferred using FastME (v2.1.6.1), a distance-based algorithm, with bootstrap values calculated from 100 replicates. Phylogenetic networks were plotted using Interactive Tree of Life browser application (https://itol.embl.de, v6.1; *Letunic and Bork, 2016*). The time to the most recent common ancestor was estimated using Network 10.0 (Fluxus-engineering, Colchester, UK) using rho values calculated by Network (*Forster et al., 1996*) and an assumed mutation rate of $0.5 \times 10^{-9}$ substitutions per base per year (*Scally and Durbin, 2012*; *Zhang et al., 2017*).

## Development of the HHV-6 Explorer

The HHV-6 Counter takes a fasta multiple alignment and compares each sequence to a selected reference HHV-6 strain to generate counts of genetic variation (substitutions, insertions, deletions) from overlapping or non-overlapping windows across the HHV-6 genome. If a corresponding GenBank file for the reference is present, HHV-6 Counter will also provide windowed counts for coding sequence and amino acid variation across the genome and for each gene. This includes missense/nonsense changes, in-frame insertions/deletions, nonsense insertions, potential splice site changes, loss of start or stop, and tracking of frameshift changes. Windows that contain non-standard base characters or have sequence gaps due to method failures are flagged but their counts are still retained in the results. HHV-6 Counter exports the count windows both as a series of Excel files and as a Python panda pickle file for use in the HHV-6 Explorer.

The HHV-6 Explorer is based on plotly Dash (https://dash.plotly.com/) and, using the output from HHV-6 Counter, allows the graphical display of variation counts for different strains across the HHV-6 genomes and/or on a per gene basis compared against a selected reference HHV-6 strain. It also displays a multiple alignment for the selected gene. A pre-populated version of HHV-6 Explorer containing the data from this manuscript can be found at https://www.hhv6explorer.org/. The source code for HHV-6 Explorer and HHV-6 Counter is available from https://github.com/colinveal/HHV6-Explorer.

## PCR primers and other oligonucleotides

The sequences of primers and hydrolysis probes used in the study are listed in *Supplementary file 7*.

## Amplification and analysis of subtelomere-iciHHV-6B junctions

Subtelomere-iciHHV-6B junctions were amplified by PCR (33 cycles) using a trial-and-error approach with primer DR8F(A/B) and various primers known to anneal to chromosomal subtelomere regions (*Zhang et al., 2017*). Amplicons were purified using a Zymoclean gel DNA recovery kit (Zymo Research, Irvine, CA) and sequenced using primer DR8FT2 or the appropriate subtelomere primer (*Supplementary file 7*). The number and pattern of telomere repeats (TTAGGG) and degenerate telomere-like repeats across the subtelomere-iciHHV-6B junction amplicons were identified and color-coded manually to generate repeat patterns.

## Amplification and analysis of HHV-6B $DR_R$-pvT1 and $DR_L$-pvT1

$DR_R$-pvT1 was amplified by PCR using primers DR1R and U100Fw2 in the first round (94 °C, 1.5 min; 25 cycles 94 °C 15 s, 62 °C 30 s, 68 °C 10 min; 68 °C 2 min) and primers DR421R and TJ1F in the second round (94 °C, 1.5 min; 25 cycles 94 °C 15 s, 64 °C 30 s, 68 °C 1.5 min; 68 °C 2 min). $DR_L$-pvT1 was amplified by STELA (see below) followed by a secondary PCR using primers DR421R and TJ1F. The short pvT1 amplicons were size-separated by electrophoresis in a 3% NuSieve (Lonza) agarose gel, extracted, and Sanger sequenced using primer TJ1F. The number and pattern of (TTAGGG) and degenerate telomere-like repeats were identified and color-coded manually to generate pvT1 repeat patterns. For comparative analysis, the $DR_R$-pvT1 repeat patterns were aligned manually by dividing them into the proximal, central, and distal regions with respect to the DR421 primer and distinctive motifs identified (*Supplementary file 3*).

## STELA and detection of newly formed telomeres at $DR_R$-T1 and $DR_L$-T2

The telomere at the end of the iciHHV-6B genome was amplified by STELA using the DR1R primer and Telorette 2/Teltail essentially as described previously (*Huang et al., 2014*; *Jeyapalan et al., 2008*). DNA was diluted to 250–1000 pg/µL for cell line DNA, 500 pg/µL for saliva DNA, 600 pg/µL for blood DNA, and 1000 pg/µL for sperm DNA. The primer concentrations in each 10 µL STELA reaction were 0.3 µM DR1R, 0.225 µM Telorette 2 and 0.05 µM Teltail. *Taq* polymerase (Kapa Biosystems) was used at 0.04 U/µL and *Pwo* (Genaxxon Bioscience, Ulm, Germany) at 0.025 U/µL. The STELA PCRs were cycled 25 times.

To detect telomeres at $DR_R$-T1, STELA was conducted as above using primers DR1R and Telorette 2/Teltail on iciHHV-6B DNA samples that showed a length difference between $DR_L$-pvT1 and $DR_R$-pvT1. Next the STELA reaction product (1 µL) was diluted 1:10 in water and used as input for PCR of pvT1 using primers DR421R and TJ1F (cycling conditions as above, 25 cycles). The amplicons were size-separated by agarose gel electrophoresis to distinguish $DR_R$-T1-associated telomeres from the majority of $DR_L$-T1-associated telomeres. The remainder of the undiluted STELA product was size-separated by agarose gel electrophoresis and amplified telomeres detected by Southern blot hybridization to a radiolabeled (TTAGGG)$_n$ probe.

To detect telomeres at $DR_L$-T2, primer UDL6R was used in STELA reactions instead of primer DR1R with 250–1000 pg genomic DNA per reaction and cycled 26 times (*Huang et al., 2014*). Amplicons hybridizing to the radiolabeled (TTAGGG)$_n$ probe that migrated at less than 900 bp were counted as unlengthened truncations, those between 900 bp and 8.6 kb were counted as lengthened truncations, and those larger than 8.6 kb were not counted as truncations. The number of truncations per cell was estimated by converting the amount of input DNA to cell equivalents on the assumption that a cell contains 6.6 pg DNA (or 3.3 pg for a haploid sperm cell) and dividing the number of truncations (lengthened and unlengthened) by the number of cells screened. To sequence $DR_L$-T2-associated telomeres, STELA products were extracted from agarose gel slices using repeated freeze-thawing and reamplified using primers DR8RT2 and Telorette 2. The recovered amplicons were Sanger sequenced.

## Detection of integrated HHV-6B in individuals with acqHHV-6B

STELA was carried out on genomic DNA from individuals with a known copy number of acqHHV-6B, measured by ddPCR. As the copy number of acqHHV-6B was low and telomere integration events expected to be lower, various precautions were introduced for these experiments. STELA reactions

were set up in a room previously unused for any STELA or HHV-6 experiments. In addition to avoid contamination with previously generated iciHHV-6A/6B STELA products, STELA primers with new barcodes (Telorette2BC28/Teltail2BC38) and a new a flanking primer (DR2RSTELA) were used. The DR2RSTELA primer anneals upstream of the DR1R primer used for other iciHHV-6A/6B STELA reactions, so preventing amplification of any previously generated STELA amplicons. STELA reactions were set up as above, but as the acqHHV-6B copy number was low in these samples, the quantity of genomic DNA added was increased to 2–5 ng per reaction and cycled 25 times as above. To test that the higher DNA input did not inhibit the potential to amplify an HHV-6B-associated telomere (if present), optimization experiments were undertaken. These included dilution of genomic DNA from an iciHHV-6B cell line (one copy per cell) with increasing quantities (0, 100, 500, 1000, and 5000 pg per PCR) of DNA from the HT1080 cell line (HHV-6A/6B negative) followed by STELA in multiple parallel reactions. This showed that neither the number of HHV-6-associated telomeres amplified by STELA nor the average length was affected in the presence of increasing quantities of human genomic DNA (up to 5 ng per reaction).

Following completion of the multiple parallel STELA reactions, 1 µL of each reaction was diluted 1:10 in water and used as input for amplification of $DR_L$-pvT1 using primers DR421R and TJ1F (cycling conditions as above, 25 cycles). The amplicons were sequenced using TJ1F to confirm that the $DR_L$-pvT1 sequence in the amplified telomere was the same HHV-6B strain as present in that saliva sample. The remainder of the undiluted STELA product was size-separated by agarose gel electrophoresis and detected by Southern blot hybridization to a radiolabeled $(TTAGGG)_n$ probe. As a control for non-specific amplification from another telomere or elsewhere in the human genome, 900 STELA reactions were conducted each with 5 ng genomic DNA from an HHV-6A/6B free cell line, HT1080 (4500 ng total). No STELA amplicons were generated from the HT1080 control reactions.

## DR circles

Double restriction digests were carried out at 37° C for 1 hr using 10 U of each enzyme (XbaI and ScaI-HF, or PstI and SacI) and 1× NEB CutSmart Buffer (New England Biolabs) in 500 µL reactions containing 10 µg 4B-11p15.5 or 1B-HHV-6B DNA. Enzymes were heat-inactivated at 80° C for 20 min. Control DNA was treated without any restriction enzymes. Diluted DNA was amplified using PCR with primers DR8F(A/B) and DR3R with a 10 min extension time (26 cycles). PCR products were size-separated on a 0.8% agarose gel with electrophoresis, and amplicons were detected by Southern blot hybridization to a radiolabeled telomere $(TTAGGG)_n$ probe.

## Statistical analysis

Data are expressed as means ± standard error of the mean (SEM). Mann–Whitney test was used to compare unpaired groups of ranked data obtained from assays conducted on DNA derived from the same biological sample type (*Figure 5C and F*). Wilcoxon test was used to compare paired groups of data obtained from treated and untreated samples. Statistical analyses were performed using Prism software (version 9.0, GraphPad Software). Outliers were not removed from data sets. Samples sizes were based on availability of suitable biological materials.

## Acknowledgements

We thank Dr Sukhvir Rai (University of Leicester) for the gift of the d37 mesenchymal stem cell line; Dr Yan Huang, Dr Enjie Zhang, Dilan Patel, and Ryan Mate for preliminary work that contributed the initiation of this study; and Drs Chiara Batini, Celia May, and Jon Wetton for their advice on phylogenetics and hypervariable markers. We thank Matthew Denniff and Charlotte Hogg for help with sample acquisition and TRAP assays, respectively. We thank staff at the HHV-6 Foundation for their continued support, and we acknowledge the contribution of members of the BIOSTAT-CHF consortium. MLW was funded by UK Biotechnology and Biological Sciences Research Council (BBSRC) and the Midlands Integrative Biosciences Training Partnership (MIBTP 1645656). The work was supported in part by the UK Medical Research Council (G0901657 to NJR and MC_UU_12014/3 to AJD); HHV-6 Foundation pilot grant to NJR; the Canadian Institutes of Health Research grants (MOP_123214) to LF. The BIOSTAT-CHF project was funded by a grant from the European Commission (FP7-242209-BIOSTAT-CHF).

## Additional information

### Funding

| Funder | Grant reference number | Author |
|---|---|---|
| Biotechnology and Biological Sciences Research Council | MIBTP 1645656 | Michael L Wood |
| Medical Research Council | G0901657 | Nicola J Royle |
| HHV-6 Foundation | Pilot grant | Nicola J Royle |
| Canadian Institutes of Health Research | MOP 123214 | Louis Flamand |
| European Commission | FP7-242209- BIOSTAT-CHF | Adriaan A Voors |
| Medical Research Council | MC_UU_12014/3 | Andrew J Davison |

The funders had no role in study design, data collection and interpretation, or the decision to submit the work for publication.

### Author contributions

Michael L Wood, Conceptualization, Formal analysis, Investigation, Methodology, Validation, Writing - original draft; Colin D Veal, Data curation, Investigation, Software, Visualization, Writing - original draft; Rita Neumann, Investigation, Validation, Writing - review and editing; Nicolás M Suárez, Formal analysis, Investigation, Validation, Writing - review and editing; Jenna Nichols, Investigation, Writing - original draft; Andrei J Parker, Diana Martin, Investigation, Resources, Writing - review and editing; Simon PR Romaine, Formal analysis, Investigation, Writing - review and editing; Veryan Codd, Nilesh J Samani, Project administration, Resources, Supervision, Writing - review and editing; Adriaan A Voors, Louis Flamand, Resources, Supervision, Writing - review and editing; Maciej Tomaszewski, Resources, Writing - review and editing; Andrew J Davison, Data curation, Investigation, Resources, Supervision, Writing - original draft, Writing - review and editing; Nicola J Royle, Conceptualization, Data curation, Funding acquisition, Methodology, Project administration, Resources, Supervision, Writing - original draft, Writing - review and editing

### Author ORCIDs

Michael L Wood http://orcid.org/0000-0003-4042-525X
Colin D Veal http://orcid.org/0000-0002-9840-2512
Andrei J Parker http://orcid.org/0000-0003-0735-4357
Andrew J Davison http://orcid.org/0000-0002-4991-9128
Nicola J Royle http://orcid.org/0000-0003-1174-6329

### Ethics

The study was conducted in accordance with the Declaration of Helsinki and with approval by the relevant ethics committees as follows: The University of Leicester's Research Ethics Committee (refs: 10553-njr-genetics; njr-61d3). The BIOSTAT-CHF study was approved by the relevant ethics committee in each centre, all participants gave their written, informed consent to participate (Voors et al, 2016).

### Decision letter and Author response

Decision letter https://doi.org/10.7554/eLife.70452.sa1
Author response https://doi.org/10.7554/eLife.70452.sa2

## Additional files

### Supplementary files

• Supplementary file 1. AcqHHV-6A/6B and iciHHV-6A/6B genomes included in study.

• Supplementary file 2. Estimated Time to Most Recent Common Ancestor (TMRCA) for carriers of iciHHV-6A and iciHHV-6B with different chromosomal locations.

• Supplementary file 3. Distinctive features of DRR-pvT1 repeat patterns associated with various

iciHHV-6B phylogenetic clades.

• Supplementary file 4. Measuring the percentage of acquired HHV-6B with a telomere, as an indicator of integration.

• Supplementary file 5. Variation in the frequency of truncations at DRL-T2 and percentage lengthened between samples.

• Supplementary file 6. Measuring the frequency of truncations at DRR-T1 in various samples.

• Supplementary file 7. Primers used in this study, including primers used to generate overlapping amplicons for iciHHV-6A/6B genome sequencing.

• Transparent reporting form

### Data availability

Sequencing data have been deposited in GenBank under accession numbers: MW049313-MW049327. The HHV6 explorer is freely available at https://www.hhv6explorer.org/ and so The source code for the HHV6 explorer and HHV6 counter are available at https://github.com/colinveal/HHV6-Explorer copy archieved at https://archive.softwareheritage.org/swh:1:rev:72136357145b0cafc0e2862a1c33c3762cb1e9a8;origin=https://github.com/colinveal/HHV6-Explorer;visit=swh:1:snp:75a2aae5391808d2f6dd68f9c7475a2689f10dc4. Other data generated or analysed during this study are included in the manuscript and supporting files.

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
