## [Decision Letter]

**Acceptance summary:**

Human Herpesvirus 6A (HHV6A) and 6B are common herpesviruses that establish lifelong infection in latent form and can cause severe disease upon reactivation. They are spread by acquired infection of free virus and by germ-line transmission of inherited chromosomally-integrated HHV-6A/6B in telomeres. The authors develop an approach to analyse a hypervariable region of the HHV-6B genome and exploit it to investigate the relationship between acquired and inherited virus, presenting evidence that HHV-6B can readily transition between telomere-integrated and free virus forms. These findings enhance our understanding of the intricate interplay between chronic viruses and their host.

**Decision letter after peer review:**

Thank you for submitting your article "Variation in Human Herpesvirus 6B telomeric integration, excision and transmission between tissues and individuals" for consideration by *eLife*. Your article has been reviewed by 3 peer reviewers, and the evaluation has been overseen by a Reviewing Editor and Sara Sawyer as the Senior Editor. The following individual involved in review of your submission has agreed to reveal their identity: Benedikt B. Kaufer (Reviewer #1).

Essential revisions:

The issues raised by the reviewers are mostly technical and require clarifications and slight modifications but not additional experiments. Overall, the manuscript is written well but some aspects in the presentation (size of the figures, naming of samples, labeling) should be improved to ease reading of this complex work. The authors should also clarify the validation of the PCR methods used to measure viral genomes and potential limitations of the approach.

*Reviewer #1 (Recommendations for the authors):*

1) Figure 1: The authors show the repeats present in the HHV-6B pvT1 region by colored boxes. The figure would immensely benefit from a legend (color box with respective repeat – as done in Figure 3C). This would make it much easier to follow the repeat types.

2) Line 199 and 204: please indicate if you are talking about Figure 2A or 2B.

3) Figure 2 —figure supplement 1 also indicates that mutations arose in U12. Why is this not discussed in the text? It might also help to separate the panes (by A, B…). In addition, the font sizes is very hard to read, even after zooming in. Please improve readability.

4) Figure 3A: The authors should highlight the shared DRR-pvT1 repeat pattern of individuals with 9q iciHHV-6B and 17p iciHHV-6B (e.g. using stars or arrows).

5) Figure 4C is not cited in the text. Please ensure that all figure panels are cited.

6) The authors discovered telomeric integration in the saliva, which represents mostly lytic replication. Did the authors also try this approach using bone marrow samples or other tissues. This would be very exciting data and I would assume that the ratio would be closer to 2:1.

7) How old was the G4P1 son? Was he an infant/baby making it more likely that the isolated DNA in the saliva was somehow derived from the mother e.g. during nursing.

*Reviewer #2 (Recommendations for the authors):*

– The relationship between different sequenced virus samples is determined based on similarity between the sequence of repeating elements with the DRR-pvT1 region. While it seems to be consistent, no statistical analysis or thresholds for similarity is shown. This can be a powerful tool if precise methods are established.

– The left-to-right terminology chosen for the different parts of the DRR-pvT1 regions may create confusion as it conflicts with the direction of the DR-L and DR-R terminology.

– In figures 4-6 The authors use the rate of presence or absence of amplified iciHHV-6 telomere ends in multiple parallel PCR reactions to determine the rate of telomere integration, excision, and elongation. It is unclear what exactly each lane contains, whether it is technical repeats of the initial STELA PCR, or the nested PCR of diluted STELA. If the latter is true, it is unclear how can this method be used to directly quantify integration, as the authors report doing. Furthermore, this test uses the absence of a product as a measured results but does not contain an internal positive control, which makes it lacking.

The expected band sizes should also be described more clearly, in the figures and the text. For example, why is there a single size product in 5D and not multiple sizes like in 5B? Where are the short products in figure 5E? What are all the bands in 6B?

– Different samples are repeating in different sections of the manuscript under different contexts, and in relation to different other samples, and are not easily identified. The connections between the different sections and the samples and data presented should be made more clear.

– The methods section about identification of iciHHV-6 samples, does not specify which test was performed for BIOSTAT-CHF samples (was it sequencing?). This section does mention a test for relationship between iciHHV-6A/6B carrier status and heart failure (line 520) which is not discussed elsewhere.

– Figure 2 —figure supplement 1 shows screenshots from the HHV-6 explorer made by the authors. While this may be valuable as a guide instructing how to use the explorer, the data itself is not clear since the text font is very small. The conclusions are difficult to judge from the figure as is.

– Figure 5—figure supplement 1 A/B/C should be referred separately in the relevant places.

– Intermediate STELA products are shown and presented before they are addressed directly and explained.

– Some data is not shown- line 353 sequencing results, line 388 ddPCR copies per cell data.

*Reviewer #3 (Recommendations for the authors):*

1. Could some of the repeat variation arise due to PCR error and sequencing artifact?

2. Is there any evidence for more than one strain per individual, and if so, how would this impact the analyses?

3. How do you distinguish viral genomic DNA from other forms of cell-free DNA that includes partial viral genomic fragments. Could FISH be used to confirm viral genome integrity?

4. What is the explanation for the increase frequency of telomere integration in LCL and pluripotent cell line, and can this be experimentally tested?

---

## [Author Response]

Essential revisions:The issues raised by the reviewers are mostly technical and require clarifications and slight modifications but not additional experiments. Overall, the manuscript is written well but some aspects in the presentation (size of the figures, naming of samples, labeling) should be improved to ease reading of this complex work.

We have replaced some figures or panels in figures (as outlined below) and we believe this has improved the presentation and legibility. It is more difficult to change the sample names as many appear in other publications (as indicated in Supplementary Table 1). Nevertheless, we understand that the number and style of sample names makes it difficult to follow some aspects of the paper. To address this we have expanded Supplementary Table 1 to included information about the type of sample, and we have listed the figures to which each sample contributes.

The authors should also clarify the validation of the PCR methods used to measure viral genomes and potential limitations of the approach.

HHV-6B copy number was determined by a duplexed ddPCR assay using the PCR amplicons with hydrolysis probe detection for HHV-6B POL (U38) and human RPP30 (human gene with 2 copies per diploid cell) as published previously (Bell *et al.,* 2014). We optimised the method in our own laboratory using an iciHHV-6 cell line known to contain a single full-length copy of the viral genome (U_1_DR_2_); the HT1080 cell line without the viral genome and with water-only no-template controls and found the published method was also robust in our hands. We have expanded the explanation in the material and methods (lines 570 -591). We used this duplexed ddPCR assay to quantify the abundance of HHV-6B in saliva samples and by using triplicate ddPCR assays on each sample we increased the sensitivity and accuracy, as indicated in the Materials and methods and shown in Figure 4A (8/52 saliva (SAL) samples were negative for HHV-6B in all three replica ddPCR assays). We subsequently confirmed the presence of HHV-6B in 42/52 of these saliva samples using the nested PCR DR_R_-pvT1 assay, which is specific for HHV-6B. It is possible that HHV-6A may also be present in some of these samples, which is a potential limitation but this does not affect the conclusions we have drawn from the HHV-6B data presented.

As required, we also used the published HHV-6B DR6B, POL (U38) and human RPP30 hydrolysis probe ddPCR assays to establish the copy number of different regions of the HHV-6B genome in iciHHV-6B positive samples. We have edited the material and methods to make this clearer (Lines 594 to 599).

We developed a novel ddPCR assay to detect HHV-6B PAC1 sequences using an intercalating dye, Evagreen. We have expanded the description of the assay in the material and methods (Lines 600- 613) and included a statement about optimisation. We used this assay to determine the ratio of DR: PAC1 for samples with acqHHV-6B samples (Figure 4E).STELA is a well establish PCR-based method that has been used extensively to amplify full-length telomeres from specific chromosome ends. Telomere amplification by STELA is dependent on the presence of 3’ single stranded (TTAGGG)n overhang, which are essential features of telomeres. The chromosome specificity is determined by the primer located in subtelomeric sequences immediately adjacent to the telomeric repeats. We have added an outline of the STELA method (page 9 lines 255-261) and two references that outline the STELA method as it was first described (Baird *et al.,* 2003) and an example of how we have used the method (Jeyapalan *et al.,* 2008). STELA has been used to measure the average telomere length on a particular chromosome end in cell lines, human samples from various tissues, and to detect and measure the abundance of very short telomeres present as cells approach senescence or crisis. Previously, we adapted STELA to amplify full-length telomeres on the end of iciHHV-6A or -6B genomes, irrespective of the chromosome location, by anchoring one primer in the DR region (typically DR1) and we validated this approach using cell lines with and without iciHHV-6A/6B; with various control primers; and by demonstrating that the telomere on the end of the viral genome shortened at the same rate (base pairs per cell division) as other telomeres within the cell line (Huang *et al.*, 2014). In this study we used STELA to verify the presence of iciHHV-6A/6B-associated telomeres for all the iciHHV-6A/6B samples available to us.

New to this study, we adapted the STELA approach to detect HHV-6B-associated telomeres in samples from individuals with low levels of acquired HHV-6B (acqHHV-6B). The problems we had to overcome were: the sensitivity required to detect rare molecules of HHV-6B (low abundance of acqHHV-6B is shown in Figure 4A) that may or may not have an associated telomere in the background of the human genomic DNA; the efficiency of the STELA method (it is less efficient than PCR amplification of short amplicons); the risk of contamination from previously generated HHV-6-STELA amplicons from other samples; mis-priming by the primers required for HHV-6B-STELA elsewhere the human genome (including all 92 telomeres); and mis-priming from within the HHV-6B genome, for example from T1. We took steps to control for artefacts and to optimise the assay as outlined and expanded in the Results section (lines 263-288) and in the Materials and methods (lines 734 -756).

Previously, we also demonstrated in two iciHHV-6B cell lines that we could detect very short telomeres at DR_L_-T2 arising from the loss of the terminal DR, T1 region and associated telomere. We validated the approach using Exo1 digestion to remove single-strand overhangs from genomic DNA. This demonstrated that STELA products could be amplified from: the end of the full length viral genome; at the DR_L_-T2 internal truncation site; and from another telomere 12q only when the single strand overhangs were present (i.e genomic DNA without Exo1 digestion) (Huang *et al.*, 2014). In the present manuscript we used the same approach to detect telomeres at DR_L_-T2 but for the first time in vivo. We showed that the frequency of telomere formation at DR_L_-T2 varied between individuals and tissues. We also measured the frequency of lengthening of DR_L_-T2 telomeres. We investigated the structure of the lengthened molecules (1) by Southern hybridisation to show the absence of DR3 and DR8 (regions within the DRs) and (2) by Sanger sequencing, which demonstrated they contain only (TTAGGG)_n_ repeats. It was not possible to sequence across the entirety of DR_L_-T2 telomeres that had been lengthened by several kilobases due to their repetitive nature. Nevertheless, sequence obtained from each end of very long DR_L_-T2 telomeres supported our conclusion that they are terminated by long arrays of (TTAGGG)_n_. While this is a limitation, we believe the combination of the sequence and Southern blot analyses support our proposal that lengthening of the newly formed DR_L_-T2 telomeres likely occurs in cells with telomerase activity. Moreover, we demonstrate (Figure 5) that the proportion of lengthened telomeres at the DR_L_-T2 varies considerably between tissues/ cell types.

To detect truncations/new telomeres at the internal DR_R_-T1 site it was necessary to distinguish between STELA amplicons that originate from the full-length iciHHV-6B genome present in most cells and the infrequent STELA amplicons that arise from a truncation at DR_R_-T1. To overcome this we identified iciHHV-6B samples that showed a size difference between the pvT1 regions in DR_R_-T1 and DR_L_-T1. This prior knowledge, combined with STELA using the DR1R primer and nested amplification of pvT1, enabled us to detect STELA amplicons that had arisen from a telomere at DR_R_-T1. We validated the approach by isolating and sequencing some pvT1 amplicons and demonstrated the pattern of repeats was as expected for DR_R_ or DR_L_ in the sample. This laborious validation was conducted on a minority of the DR_R_-pvT1 and DR­_L_-pvT1 amplicons but included some reactions where both bands were present (gel slices were carefully excised to separate the two pvT1 bands prior to Sanger sequencing). The detection of truncations at DR_R_-T1 is limited to samples where there is prior knowledge of a length difference between the pvT1 in DR_R_ and DR_L_. In addition, the size difference between the amplicons is usually small, influencing the ability to size separate the amplicons. Consequently, only iciHHV-6B samples with large DR_R_ vs DR_L_ pvT1 size differences, were selected for analysis. We have added a statement in the Results section to address this point (lines 362-373) and referred to it again in the discussion (line 510).

Reviewer #1 (Recommendations for the authors):1) Figure 1: The authors show the repeats present in the HHV-6B pvT1 region by colored boxes. The figure would immensely benefit from a legend (color box with respective repeat – as done in Figure 3C). This would make it much easier to follow the repeat types.

We have added a colour key for the TTAGGG and the degenerate telomere-like repeats present in the pvT1 amplicons to Figure 1; and Figure 3 and adjusted the legends accordingly. There is relatively little overlap between the variety of degenerate telomere-like repeats identified in the subtelomere-HHV-6B junctions and the degenerate telomere-like repeats present in pvT1 therefore the colour coding keys are different and consequently Figure 3 now has two keys.

2) Line 199 and 204: please indicate if you are talking about Figure 2A or 2B.

We have now stated which Figure 2 panels are being referred to in this section (Figure 2A in line 199; Figure 2B in line 204) and agree this aids clarity in this paragraph.

3) Figure 2 —figure supplement 1 also indicates that mutations arose in U12. Why is this not discussed in the text? It might also help to separate the panes (by A, B…). In addition, the font sizes is very hard to read, even after zooming in. Please improve readability.

With respect to the mutation in HHV-6B U12 – it is present in Z29 and is documented in the Genbank record (https://www.ncbi.nlm.nih.gov/nuccore/AF157706) whereas U12 is full length in the HST Genbank record (https://www.ncbi.nlm.nih.gov/nuccore/AB021506).

The point we want to illustrate in Figure 2 —figure supplement 1 is that inactivating mutations have arising in iciHHV-6A and -6B genomes after integration. As the Z29 HHV-6B genome is quite diverged from most of the iciHHV-6B samples in the currently available phylogenetic trees (for example Figure 3B) and the mutation in U12 is not relevant to the point we wanted to make we have revised the images captured form HHV-6 Explorer and not included comparison with the Z29 genome.

We apologise for the low resolution of the images in Figure 2 —figure supplement 1. To address this we have separated the images for HHV-6A and HHV-6B across two panes as suggested. Some of the text extracted directly from the HHV-6 explorer has been overwritten to increase the font size and make it legible.

4) Figure 3A: The authors should highlight the shared DRR-pvT1 repeat pattern of individuals with 9q iciHHV-6B and 17p iciHHV-6B (e.g. using stars or arrows).

We have done this and adjusted the figure legend accordingly.

5) Figure 4C is not cited in the text. Please ensure that all figure panels are cited.

This panel was mentioned at the same point as Figure 4B but to ensure it is not so easily overlooked we have edited line 269 to: (Figure 4B, 4C). We have also checked that all the other figure panels are cited in the manuscript.

6) The authors discovered telomeric integration in the saliva, which represents mostly lytic replication. Did the authors also try this approach using bone marrow samples or other tissues. This would be very exciting data and I would assume that the ratio would be closer to 2:1.

Our primary aim for this part of the study was to see if we could gather evidence indicative of telomeric integration in vivo. First, we needed to determine whether the donors had been exposed to the virus at some point and then move to detection of HHV-6B-associated telomeres. For these reasons we collected saliva samples from healthy adult volunteers. We have not had access to other tissues, such as bone marrow, but it would be interesting to investigate. We agree that the ratio of telomere-integrated to free virus is likely to vary between tissues that harbour latent virus.

7) How old was the G4P1 son? Was he an infant/baby making it more likely that the isolated DNA in the saliva was somehow derived from the mother e.g. during nursing.

This is an interesting point but the children in the families were 18 years or older. G4P1 was 28.8 years at the time of sample collection. We have added the ages of the family members to the pedigrees in Figure 7.

Reviewer #2 (Recommendations for the authors):– The relationship between different sequenced virus samples is determined based on similarity between the sequence of repeating elements with the DRR-pvT1 region. While it seems to be consistent, no statistical analysis or thresholds for similarity is shown. This can be a powerful tool if precise methods are established.

pvT1 is a complex repeat region in terms of the variety of degenerate repeats, the interspersed nature of different repeats and the differences in repeat number. As far as we are aware, there are no available computer programs that align these types of repetitive sequences accurately. Previous work on tandem repeat sequences (our work on human telomeres and other groups on complex VNTRs (Variable Number of Tandem Repeats)) has shown that manual alignment can be used successfully to make comparisons. For example, in this work manual alignment allowed us to see immediately that the pvT1 distal region is the least variable, although not completely static (see Figure 1—figure supplement 1). Furthermore, pvT1 in iciHHV-6B and acqHHV-6B will mutate at different rates but have interconnected though different evolutionary histories, influenced by viral genome integration and excision rates (accompanied by full reactivation) that are currently unknown. These complicating features along with a limited number of pvT1 repeat patterns from some iciHHV-6B clades (shared integration sites) has limited our ability to set similarity thresholds or use statistical analysis with the data available. Nevertheless, we recognise the importance of the point raised here and so to address this point we have listed the distinctive features of DR_R_-pvT1 repeat maps among iciHHV-6B samples in the same clade (new Supplementary Table 3) and adjusted the text of the manuscript (lines 221-226). Formulation of the distinctive features was informed by known relationships between iciHHV-6B genomes based on whole genome sequence similarity, subtelomere-iciHHV-6B junctions, and FISH. In addition, the presence of a particular motif or variant repeat in both DR_R_-pvT1 and DR_L_-pvT1 in an individual iciHHV-6B genome indicated that it was present at integration, and this was a useful aid to defining distinctive features. We also include a statements that discuss how these pvT1 repeat patterns were compared and the desirability of an automated approach (lines 447-473).

– The left-to-right terminology chosen for the different parts of the DRR-pvT1 regions may create confusion as it conflicts with the direction of the DR-L and DR-R terminology.

We see how confusion could arise so we have change the labels for the pvT1 sections to proximal, central and distal with respect to the DR421R primer used in nested PCR to amplify pvT1 from either DR_R_ or DR_L_. The changes have been made to Figure 1, the manuscript (Lines 148-151) and the Figure 1 legend (lines 1134-1136).

– In figures 4-6 The authors use the rate of presence or absence of amplified iciHHV-6 telomere ends in multiple parallel PCR reactions to determine the rate of telomere integration, excision, and elongation. It is unclear what exactly each lane contains, whether it is technical repeats of the initial STELA PCR, or the nested PCR of diluted STELA. If the latter is true, it is unclear how can this method be used to directly quantify integration, as the authors report doing.

We apologise that our explanations about how STELA was used are unclear. In brief, each STELA PCR is conducted on a very small aliquot of human DNA as this limits the number of ‘target’ molecules that could potentially be amplified. For STELA of a human telomere with two copies per cell the quantity of genomic DNA added to each PCR is typically 250-1000 pg (38-152 cell equivalents). Several parallel reactions (typically 5-10) are set up and the data collect across the reactions pooled for subsequent comparative or statistical analysis. If the quantity of genomic DNA added to each STELA is increased to 10 ng per reaction (typical input for a normal PCR) many telomeres are amplified and the pooled products are detected as a smear on the Southern blot.

In Figure 4, to detect HHV-6B-associated telomeres (if present) in the samples from individuals with low levels of acqHHV-6B (average 0.00145 copies per cell, line 252) we increased the quantity of DNA per STELA PCR. To establish how much DNA could be added per STELA PCR without inhibiting amplification from an HHV-6B-associated telomeres (if present) we conducted doping experiments by adding genomic DNA from an iciHHV-6B cell line (one copy per cell) with increasing quantities of DNA from the HT1080 cell line (HHV-6A/6B negative) followed by STELA in multiple parallel reactions. From these experiments we concluded that 2-5ng genomic DNA from a person with acqHHV-6B per STELA PCR was optimal.

In Figure 5, there was a different set of considerations when detecting truncations (with or without lengthening) at DR_L_-T2 but as we found the frequency of truncations was high, 250-1000 pg genomic DNA was added to each STELA PCR. To isolate individual short DR_L_-T2 telomeres for sequencing STELA PCRs were set up with lower quantities of DNA.

In Figure 6, detection of truncations and new telomere formation at DR_R_-T1 could not be achieved using STELA alone as the primers used can amplify telomeres present at DR_L_-T1 (the end of the full length iciHHV-6B genome) and new telomeres formed at the internal DR_R_-T1 following truncation here. Consequently, in these experiments STELA was used to amplify telomeres from both sites and subsequent analysis of the hypervariable pvT1 region was used to distinguish the origin of the products. A limitation of this approach is that it can only be conducted on sample that shown length variation between the pvT1 region in DR_R_ and DR_L_.

We have made several changes to address these points raised by Reviewer 2. First, we have included a brief description of the STELA method that highlights how it differs from ‘regular’ PCR and then explained how STELA PCRs were set up to detect HHV-6B-associated telomeres in acqHHV-6B samples (lines 255-272). We have expanded a sentence to indicate that the acqHHV-6B integration frequencies should be interpreted cautiously (lines 286-288). We have also expanded our explanation of the method used, precautions taken and outlined optimisation experiments that were undertaken (lines 731 -756).

Similarly, we amended the text with regard to detection of truncations and newly formed telomeres at DR_L_-T2 (lines 306-308) and DR_R_-T1 (lines 3602-373) and indicate a limitation of the approach to detect truncations at DR_R_-T1.

We have amended the legends for figures 4, 5 and 6 so that it is clearer what is shown in the Southern blots from the multiple parallel STELA reactions.

Figure 4 legend: page 34,

Figure 5 legend: page 35-36,

Figure 6 legend: page 36.

Furthermore, this test uses the absence of a product as a measured results but does not contain an internal positive control, which makes it lacking.

Reviewer 2 has raised an important point about how to interpret STELA PCRs that are negative. Only the assay used to detect HHV-6B-associated telomeres in samples from individuals with acqHHV-6B (Figure 4) resulted in a majority of negative tests and so here we address the point raised with respect to that assay. During the development of the assay we considered this issue carefully but we were unable to devise a suitable internal PCR control that would be meaningful and yet not compromise the amplification for acqHHV-6B-associated telomeres, if present. We also predicted that acqHHV-6B-associated telomeres could be long and vary in length between molecules. A control amplicon (of any size) within the human genome would be present at a copy number that massively exceeded the copy number of acqHHV-6B within the sample and therefore was unsuitable. An alternative would be to design a control amplicon within the viral genome and use it in duplex PCRs however there was no way to test whether there would be interference between the pairs of primers or to establish the optimum cycling conditions. Importantly, if the control amplicon was shorter than an acqHHV-6B-associated telomere it would amplify more efficiently and out compete an acqHHV-6B-associated telomere amplicon (if present). Instead of including an internal control, we took steps to ensure that all the STELA PCRs were equivalent, whether or not an amplicon is generated. Consequently, the multiple parallel PCRs were set up as a large volume ‘mastermix’ that comprised all the reagents required for PCR and the genomic DNA from the selected donor. Following thorough mixing the master mix was aliquoted into 90 individual wells of 96 well PCR plates (10 ul per well) so partitioning the DNA into hundreds of parallel reactions that were scored separately. Consequently, generation of an amplicon in a small number of the PCRs partially serves as a control. In addition, as a positive control for the whole PCR run, a small volume of the ‘mastermix’ was taken out (before adding donor DNA) and used to make up six control reactions (per 96 well plate) with added iciHHV-6B genomic DNA. To calculate the percentage of acqHHV-6B genome with a telomere we used the number of STELA PCRs that were positive (generated an amplicon that hybridised to the radiolabelled (TTAGGG)_n_ probe). Our estimations of the percentage of acqHHV-6B genomes with a telomere (Figure 4D and Sup Table 4) assume that a negative test did not contain a copy of acqHHV-6B with a telomere rather than such a molecule was present but failed to amplify. If this assumption is untrue for a proportion of the negative reactions the estimation acqHHV-6B genomes with a telomere will be underestimated but unfortunately it is not possible determine the error rate with the data available at this time.

We believe we addressed the uncertainties with respect to the frequencies of integrated acqHHV-6B genomes (page 9 lines 283-286) but in response to the point raised we have emphasised the need for caution by adding the sentence: ‘For these reasons the error rate associated with detection of acqHHV-6B-associated telomere is unknown and the integration frequencies should be interpreted cautiously.’ (page 9, lines 286-287).

The expected band sizes should also be described more clearly, in the figures and the text. For example, why is there a single size product in 5D and not multiple sizes like in 5B? Where are the short products in figure 5E? What are all the bands in 6B?

We have edited the manuscript and Figure 5 legend to add information about the predicted size ranges of the novel telomere are DRL-T2 (page 10 lines 306-308; already stated line 333; and Figure 5 legend page 35, lines 1233-1234). The Southern blots in Figure 5E show the analysis of blood DNA from an iciHHV-6B carrier (401027) in which 63.7% of the novel telomeres at DR_L_-T2 were lengthened (Supplementary Table 5). Given the high frequency of lengthened telomeres at DR_L_-T2 in this sample, it was just chance that short telomeres (in the 0.7-0.9kb range) were not detected in the reactions shown in these blots. Furthermore, in the original Figure 5E the Southern blot images were shown as cropped images because lower parts were blank. To address this point we have replaced the cropped images in Figure 5E with the full length blots and added the yellow and mauve lines to mark where short telomere at DRL-T2 would appear, if present.

Figure 6B demonstrates that we can also detect truncations and novel telomere formation at the internal DR_R_-T1. As outlined in Figure 6A and above, STELA with the flanking DR1R primer amplifies telomeres from full length copies of the iciHHV-6B genome (present at approximately 1 copy per cell) as well as novel telomeres formed at DR_R_-T1. Discrimination between the two types of amplicons can be achieved in samples where there is a length difference between pvT1 in DR_R_ and DR_L_. In Figure 6B, the upper STELA Southern blot images show amplification and detection of telomeres from the end of full-length viral genome and small number of novel telomeres at DR_R_-T1. The lower ethidium bromide stained gel images show the secondary nested PCR of pvT1 from each of the STELA reactions. The reactions that contain an amplified pvT1 fragment from DR_R_-T1 are identified with a white dot. Figure 6 legend (lines 1257- 1274) has been edited to explain more clearly what is shown.

– Different samples are repeating in different sections of the manuscript under different contexts, and in relation to different other samples, and are not easily identified. The connections between the different sections and the samples and data presented should be made more clear.

As indicated above, many of the samples have appeared in other publications and so it was not possible to simplify the sample names. However, to address this point we have added two columns to the Supplementary table 1. These columns indicated the sample type (e.g. saliva, blood or cell line origin) and if the samples have been used to contribute to the data in the various figures.

– The methods section about identification of iciHHV-6 samples, does not specify which test was performed for BIOSTAT-CHF samples (was it sequencing?). This section does mention a test for relationship between iciHHV-6A/6B carrier status and heart failure (line 520) which is not discussed elsewhere.

The iciHHV-6 positive sample in the BIOSTAT-CHF cohort were identified by a duplexed screen that included a positive PCR control amplicon in the human genome and an amplicon to detect DR3 in HHV-6A or 6B. The positive samples were then verified with various other HHV-6A/6B amplicons. A more complete description of the screen has been included in the materials and method (lines 557-561).

We did not find any statistically significant relationship between iciHHV-6A/6B status and the markers of heart failure that could be investigated in the BIOSTAT-CHF cohort. However, we only identified 19 iciHHV-6A or -6B carriers in the cohort, which limits the power to detect associations if present. We have not included the analysis of the iciHHV-6 status in the BIOSTAT-CHF cohort as it does not fit with the overall aim of the investigations presented, which focus on the relationship between acqHHV-6B, iciHHV-6B strains and their dynamic relationships with human telomeres.

– Figure 2 —figure supplement 1 shows screenshots from the HHV-6 explorer made by the authors. While this may be valuable as a guide instructing how to use the explorer, the data itself is not clear since the text font is very small. The conclusions are difficult to judge from the figure as is.

We note that reviewer 1 also commented on the poor resolution of these images. As indicated above the Figure 2—figure supplement 1 has been split in to two panels (HHV-6A and HHV-6B) and the images captured from the HHV-6 Explorer now have a higher resolution. In addition, some text has been overwritten to increase the font size.

– Figure 5—figure supplement 1 A/B/C should be referred separately in the relevant places.

This has been amended. Please see: Figure 5—figure supplement 1 A lines 315/316; Figure 5—figure supplement 1 B line 339; Figure 5—figure supplement 1 C line 340.

– Intermediate STELA products are shown and presented before they are addressed directly and explained.

We note this point. The intermediate length STELA products are visible in Figure 5B, with Figure 5B first referred to line on 310 and they are discussed in detail the same section of the manuscript (lines 330-348). We have not edited the text to address this point as we believe it would disrupt the flow of the manuscript.

– Some data is not shown- line 353 sequencing results,

in Figure 6B expected pattern of repeats for DR_R_-pvT1 and DR_L_-pvT1 are shown below the photograph of the gel images. The pvT1 repeat patterns were obtained from Sanger sequencing traces and match this shown in Figure 6B. The text (lines 368-369) has been amended for clarification.

line 388 ddPCR copies per cell data.

the HHV-6B ddPCR copy number data that accompany Figure 7 are now shown in a new Figure 7—figure supplement 1, and referred to in the manuscript (line 408).

Reviewer #3 (Recommendations for the authors):1. Could some of the repeat variation arise due to PCR error and sequencing artifact?

This is extremely unlikely. We routinely PCR amplify repetitive sequences and find no issue with reproducibility if the PCR conditions are optimised and the cycle number kept low to avoid generation of ‘collapsed’ PCR artefacts. Furthermore, we do not clone the amplified products so the chance of generating a PCR induced Sanger sequence artefact is almost non-existent, unless it arises in the first one or two cycles. A good example of the reproducibility of the pvT1 assay is shown by our analysis of DR_R_-pvT1 in the three generations of the CEPH 1375 family (Figure 1C). Also, for many samples we have amplified the pvT1 regions in two or more independent reactions and reproducibly generate the same length product and pvT1 repeat map.

2. Is there any evidence for more than one strain per individual, and if so, how would this impact the analyses?

We did not find any evidence of two different copies of acqHHV-6B in the human samples screened. As the copy number of acqHHV-6B was low, a second strain would have to be present at similar level and contain a DR_R_-pvT1 sequence with a different length to be detected. In summary our study was not set up to address this question and we cannot exclude the hypothetical possibility that more than one strain could be present.

While we agree that this is an interesting question, we do not see that the presence of another acqHHV-6B strain present at a very low level would affect the conclusions we have drawn. For example, a second strain of acqHHV-6B would be included in our copy number analysis by ddPCR and would also have potential to integrate and form de novo telomeres, therefore our estimated proportion of copies of HHV-6B that are integrated would not be affected.

3. How do you distinguish viral genomic DNA from other forms of cell-free DNA that includes partial viral genomic fragments. Could FISH be used to confirm viral genome integrity?

We extract ‘bulk’ high molecular weight genomic DNA from the samples collected so we cannot categorically distinguish between various types of DNA. However, cell free DNA is fragmented (average length 167 nucleotides) and often single stranded, and both these features are likely to reduce the efficiency of recovering it in the extraction methods used. Furthermore, the short size of cell free DNA will prevent it from contributing to most assays presented.

We are not confident we understand the point raised about using FISH to confirm viral genome integrity. If it is with respect to determining whether an iciHHV-6B carrier has a full viral genome or DR-only, then yes FISH with appropriate probes could be used. But more simply and cheaply we used a PCR base approached for this.

4. What is the explanation for the increase frequency of telomere integration in LCL and pluripotent cell line, and can this be experimentally tested?

With respect to this point we believe Reviewer 3 is referring to the data shown in Figure 5, which explores the frequencies of partial excisions (truncations) that result in loss of the terminal DR_L_ and new telomere formation at DRL-T2. It is unclear at the moment why partial excision events are more frequent in LCLs and pluripotent cell lines. It could be related to chromatin organisation, which we attempted to address with the TSA analysis (Figure 5D). It could also be related to the chance of ‘capturing’ or ‘retaining’ a cell that has undergone an excision. A cell that has one or a few critically short telomeres is likely to exit from the cell cycle (senesce) but such cells may persist for a longer time in culture (e.g. LCLs). Notably the pluripotent cells had a high frequency of the partial excision, but also a high proportion of the new telomere were lengthened and this would allow the cell to survive.

Yes, there are some exciting possibilities to test various hypotheses, although neither LCLs nor pluripotent cells are easy to manipulate, for example by gene editing approaches.